# A Tale of Two Symmetries: Exploring the Loss Landscape of Equivariant Models

**YuQing Xie**
Massachusetts Institute of Technology
Cambridge, MA 02139
xyuqing@mit.edu

**Tess Smidt**
Massachusetts Institute of Technology
Cambridge, MA 02139
tsmidt@mit.edu

## Abstract

Equivariant neural networks have proven to be effective for tasks with known underlying symmetries. However, optimizing equivariant networks can be tricky and best training practices are less established than for standard networks. In particular, recent works have found small training benefits from relaxing equivariance constraints. This raises the question: do equivariance constraints introduce fundamental obstacles to optimization? Or do they simply require different hyperparameter tuning? In this work, we investigate this question through a theoretical analysis of the loss landscape geometry. We focus on networks built using permutation representations, which we can view as a subset of unconstrained MLPs. Importantly, we show that the parameter symmetries of the unconstrained model has nontrivial effects on the loss landscape of the equivariant subspace and under certain conditions can provably prevent learning of the global minima. Further, we empirically demonstrate in such cases, relaxing to an unconstrained MLP can sometimes solve the issue. Interestingly, the weights eventually found via relaxation corresponds to a different choice of group representation in the hidden layer. From this, we draw 3 key takeaways. (1) By viewing the unconstrained version of an architecture, we can uncover hidden parameter symmetries which were broken by choice of constraint enforcement (2) Hidden symmetries give important insights on loss landscapes and can induce critical points and even minima (3) Hidden symmetry induced minima can sometimes be escaped by constraint relaxation and we observe the network jumps to a different choice of constraint enforcement. Effective equivariance relaxation may require rethinking the fixed choice of group representation in the hidden layers.

## 1 Introduction

*It was the best of models, it was the worst of models, it was the age of augmentation, it was the age of equivariance...*

Many domains possess latent symmetries, inspiring the development of equivariant networks which by design already account for these symmetries [12, 13, 47, 53, 54]. Such networks have proven to be extremely effective in a wide variety of tasks including molecular force fields [6, 7, 37], catalyst discovery [35], charge density prediction [23], generative models [27], protein structure prediction [29, 33], and particle physics [9, 10]. One of the major benefits of equivariance is improved sample complexity, and better generalizability, both of which have concrete theoretical support [8, 19, 36].

However, it also has been observed that equivariant networks can be tricky to optimize and best training practices for equivariant networks are less established [30, 35, 51]. In addition, there have been a number of works exploring the relaxation of equivariance constraints [43, 49, 50, 52]. The motivation for these works is model misspecification. Real world data may not have the perfect

39th Conference on Neural Information Processing Systems (NeurIPS 2025).

symmetry assumed in equivariant models, so it could even be harmful to have perfect equivariance. Indeed, relaxed models do appear to have a small performance benefit over perfectly equivariant models [49, 52] and can have better theoretical performance on such data [43, 50].

Combining these observations, a natural question arises, can relaxing equivariance improve training even when we expect a perfectly equivariant final solution? This question is addressed in [41]. In this work, an additional unconstrained linear contribution was added to the equivariant layers and regularization terms were added to the loss to still encourage equivariance. At the end of training, this additional contribution is removed, projecting the model back to an equivariant subspace. Empirically, they demonstrate this procedure improves training performance on a variety of equivariant models. While compelling, the improvements are often small and there is little theoretical insight on how specifically the relaxation helps.

In our work, we seek to address an even more foundational question: **Do equivariance constraints themselves create fundamental obstacles to optimization?** In this paper, we show that our choice of constraint enforcement can hide parameter symmetries of the larger unconstrained model class and that these hidden symmetries can induce critical points and even spurious minima.

We structure our paper as follows. In Section 2, we introduce the key concepts related to equivariance and parameter symmetry needed for our work. In Section 3 we apply these concepts to analyze the loss landscape. In particular, we prove that parameter symmetries of models in an unconstrained function space can create critical points and even spurious minima for the constrained model. We then apply these findings to equivariant networks and characterize their hidden fixed point subspaces. In Section 4, we empirically explore loss landscapes in a teacher-student setup. We validate our theoretical findings of spurious local minima and empirically demonstrate the benefits of relaxation. In particular, the relaxed model chooses a different group representation for its hidden layer.

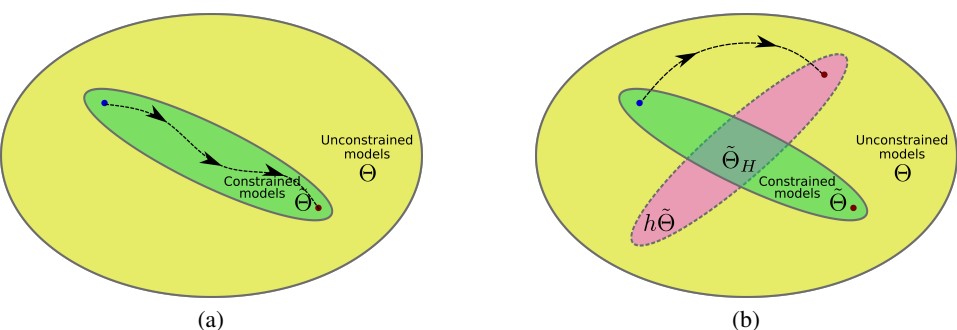

(a)                                        (b)

Figure 1: (a) Standard viewpoint of why constraints may help. If we know the ground truth function satisfies some type of constraint, then enforcing such constraints may reduce the search space allowing for better training. (b) In reality, unconstrained networks contain many parameter symmetries. However, enforcing constraints (such as equivariance) require choices that breaks some of these symmetries. These broken symmetries are "hidden" to the constrained model but still influence gradient values and in some cases create spurious minima and hinder learning. Further, we empirically observe that when completely relaxing equivariance constraints, a network can sometimes "jump" between subspaces corresponding to different ways of enforcing the same type of constraint.

As a result, we find three key takeaways from our work:

- By viewing a given architecture as a constrained version of a larger function class, we can uncover previously hidden parameter symmetries that were broken by our choice of constraint enforcement.

- Hidden symmetries have important consequences on loss landscape and can induce critical points and even minima.

- Empirically, we find a spurious minimum in an equivariantly constrained subspace can escape to a better minimum in a symmetrically related subspace through constraint relaxation. To achieve more effective relaxation of equivariant models, one may have to rethink our fixed choice of group representation in the hidden layers

## 2 Background

Here, we introduce the key concepts necessary for our work. A more in depth overview can be found in Appendix C including an overview of necessary group theory.

### 2.1 Equivariant networks

We first formally define equivariance.

**Definition 2.1** (Equivariance). Let $G$ be a group with some group action on spaces $X, Y$. A function $f : X \to Y$ is $G$-equivariant if for all $g \in G$, $x \in X$, $y \in Y$ we have $f(gx) = gf(x)$.

Essentially transforming the input is the same as transforming the output of an equivariant function. Importantly, the composition of $G$-equivariant functions is still $G$-equivariant. Hence in practice one often implements equivariant neural networks by designing equivariant layers and composing them.

#### 2.1.1 Group representations and irreps

One usually works with vector spaces and linear group actions. Such an action is referred to as a group representation.

**Definition 2.2** (Group representation). Let $G$ be a group and $V$ a vector space. A group representation is a homomorphism $\rho : G \to \mathrm{GL}(V)$.

In many cases, we can explicitly decompose representations into a direct sum of representations on smaller subspaces. Representations which cannot be decomposed further are irreps.

**Definition 2.3** (Irreducible representation). Let $G$ be a group and $V$ a vector space. A group representation $\rho : G \to \mathrm{GL}(V)$ is irreducible if there is no subspace $W \subset V$ other than 0 such that $gW = W$ for all $g \in G$.

The irreps are well understood for many groups. In particular, Schur's lemma gives us an easy way to parameterize maximally expressive equivariant linear layers once we have an irrep decomposition.

#### 2.1.2 Nonlinearities and permutation representations

There have been a number of studies analyzing the construction of equivariant networks [12, 14, 21, 26, 47, 53]. Our work focuses on networks which use permutation representations. These include group convolutions [12], Deep Sets [54], and spherical signals [13]. The reason for working with permutation representations is that they are the only type of representation compatible with arbitrary pointwise nonlinearities [39]. Such networks can be viewed as a subspace of unconstrained MLPs, which are relatively well studied.

**Definition 2.4** (Permutation representation). Let $G$ be a group. Any group homomorphism $\pi : G \to \mathrm{Sym}(n)$ is an abstract permutation representation. Let $\rho : \mathrm{Sym}(n) \to \mathrm{GL}(V_n)$ map corresponding permutation matrices acting on $n$-dimensional vector space $V_n$. Then $\rho \circ \pi : G \to \mathrm{GL}(V_n)$ is a permutation representation. We will abbreviate permutation representation as p-rep and $\rho \circ \pi$ to $\rho$.

The intuition is that for a p-rep, there is always some choice of basis $b_i$ such that $\rho(g)b_i = b_{\pi(i)}$. In other words, group actions just permute the basis. The trivial representation is an example since there is only one basis and it remains invariant under group action. The regular representation commonly used in group convolutions is another example since we can associate each basis vector to a group element [12]. In Appendix C.1, we discuss how to generalize p-reps to infinite dimensions.

Just like how irreps are building blocks of arbitrary representations, we are interested in the building blocks of p-reps. Such p-reps are the transitive p-reps.

**Definition 2.5** (Transitive permutation representation). Let $G$ be a group and $\rho : G \to \mathrm{GL}(V_n)$ be a permutation representation. We say the permutation representation is transitive if there is a basis $b_1, \ldots, b_n$ of $V_n$ such that the action of $G$ on the set $\{b_1, \ldots, b_n\}$ is transitive.

It turns out that there is a nice way to characterize all possible transitive p-reps.

**Lemma 2.6.** *Let $G$ be a group. There is a canonical bijection between the transitive permutation representations of $G$ and the classes of conjugate subgroups.*

Another commonly used nonlinearity is a tensor product. Given any pair of representations, we can always form a new representation by taking the tensor product of the vector spaces. The mapping from the pair of representations into the tensor product space is a bilinearity. Tensor products are often used in irrep based equivariant frameworks such as e3nn [26]. It turns out for many groups, there is a correspondence between pointwise nonlinearities and tensor products so our results may be relevant in those architectures as well.

## 2.2 Parameter symmetries and loss landscapes

Loss landscapes have been studied in numerous works and provides important insight on training and generalization performance [2, 11, 15, 16, 20, 22, 25, 34, 46]. Of particular interest for us is the impact of parameter symmetries on loss landscape geometry. In [11], it was shown that permutation symmetries can induce critical points creating saddles in the loss landscape. This was further explored in [46] where permutation symmetries induce entire affine subspaces of global minima subspaces (which they call an expansion manifold) and symmetry induced critical subspaces. These ideas are complementary to our work.

First, we define what we mean by parameter symmetry. In particular, we highlight that we should distinguish these into local and global parameter symmetries.

**Definition 2.7** (Parameter symmetries). Let $\mathcal{F}_\Theta = \{f_\theta : X \to Y : \theta \in \Theta\}$ be a class of functions parameterized by $\theta \in \Theta$ where $\Theta$ is the vector space of our parameters. Any homeomorphism $p : \Theta \to \Theta$ is a local parameter symmetry at $\theta$ if $f_\theta = f_{p\theta}$ and we denote the group of such transformations as $P_\theta$. If $p$ is a local parameter symmetry for all $\theta \in \Theta$, then we say $p$ is a global parameter symmetry and we denote the group of such transformations as $P$.

For example, permuting neurons in a MLP gives a global parameter symmetry as this does not change the output of the network. Permuting the output weights of two neurons is a local parameter symmetry if those two neurons share the same input weights. However, if the input weights are different, then permuting the output weights generically gives different functions so this is not a global parameter symmetry. We will focus on the effects of global parameter symmetries in this paper.

Next, given any pair of outputs, we choose a loss function $\ell : Y \times Y \to \mathbb{R}$ to measure the discrepancy. Given a distribution $D_{X,Y}$ of input-output pairs, we can use the loss function to define a loss landscape as a function $L : \Theta \to \mathbb{R}$ given by $L(\theta) = \mathbb{E}_{x,y \sim D_{X,Y}}[\ell(f_\theta(x), y)]$. Note that by construction, the loss landscape is invariant under global parameter symmetries and has the local parameter symmetries $P_\theta$ at any given $\theta$.

Of particular interest for us are spaces which are highly symmetric.

**Definition 2.8** (Fixed point subspace). Let $G$ be a group with a linear action on some vector space $V$. Let $H \leq G$ be a subgroup. Then the fixed point subspace is

$$V_H = \{v \in V : hv = v \,\forall h \in H\}.$$

For us, we usually consider $\Theta$ as our vector space and subgroups of the global parameter symmetries $H \leq P$ with a linear action. Most commonly considered parameter symmetries act linearly on parameter space.

Fixed point subspaces are useful because of the principle of symmetric criticality which has widespread usage in physics [40].

**Proposition 2.9** (Principle of symmetric criticality). *Let $H$ be a group which acts on a manifold $M$ through diffeomorphisms. Define*

$$\Sigma = \{p \in M : gp = p \,\forall g \in H\}.$$

*Let $f : M \to \mathbb{R}$ be a smooth function invariant under $H$. If $\Sigma$ is a submanifold, then critical points of $f|_\Sigma$ are critical points of $f$.*

We see that the loss landscape is invariant under $P$ and in particular, invariant under any subgroup of $P$. For example, if we consider the subgroup $H$ corresponding to permuting two specific neurons, the corresponding fixed point subspace $\Theta_H$ corresponds to shared input and output weights for those two

neurons. This is effectively a network with one fewer neuron. The principle of symmetric criticality then tells us minima of $L|_{\Theta_H}$ is a critical point of $L$, reminiscent of previously known results relating minima of smaller networks to critical points of larger ones [11, 24, 46]. To connect directly back to these prior works, we note that any transformation of the output weights of these two neurons which leaves the sum unchanged is a local symmetry of $\Theta_H$. Hence taking the orbit of $\Theta_H$ gives a larger subspace corresponding to the different ways we may embed the smaller network in a larger one.

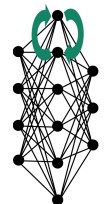  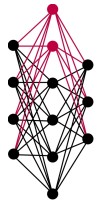  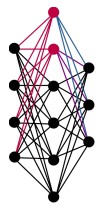  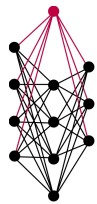

(a) Permuting neurons    (b) Fixed point subspace    (c) Reducible neuron    (d) Smaller network

Figure 2: (a) Subgroup of global parameter symmetry which permutes two neurons. (b) Corresponding fixed point subspace where both input and output weights of these two neurons are the same. (c) There is an additional local parameter symmetry on the output weights. In particular, as long as the sum of output weights of the neurons is the same, we compute the same function. (d) Smaller network with one fewer neuron computing the same function.

It will be useful to also define the "generic" regions of parameter space where we cannot reduce our network into a smaller one. If all ingoing weights of two nodes are the same, then we can merge these two nodes and create a single node with the shared ingoing weights and outgoing weights which is a sum of the outgoing weights of the original two nodes. Further, any node which has all 0 outgoing weights can be eliminated from the network. We if we cannot perform such reductions, then we say the weights are irreducible [46].

**Definition 2.10** (Irreducible neurons). We call a block of $m$-neurons in the same layer irreducible if all $m$ ingoing weights are distinct and nonzero, and the outgoing weight vectors are nonzero.

## 3 Loss landscape of equivariant networks

Here we present the main theoretical results and intuition for why they are true. Proofs can be found in Appendix D. We first make some general statements that apply to general linearly constrained models, not just equivariant ones.

### 3.1 Hidden symmetries in constrained models

The key insight is that global parameter symmetries can disappear once we restrict to constrained subspace. However, these symmetries still impact the loss landscape of the constrained model.

If the unconstrained parameter space is $\Theta$, let us denote the linearly constrained subspace by $\tilde{\Theta}$. Then of course any $p$ acting on any $\tilde{\theta} \in \tilde{\Theta}$ leaves the resulting function unchanged. However, the problem is that we may have $p\tilde{\theta} \notin \tilde{\Theta}$. Hence, such symmetries may be overlooked when analyzing the loss landscape. We call such symmetries "hidden" symmetries.

**Definition 3.1** (Hidden symmetries). Let $\mathcal{F}_\Theta$ be a class of functions parameterized by $\Theta$. Let $P$ be the group of global parameter symmetries. Let $\tilde{\Theta} \subset \Theta$ be a subspace of parameters satisfying some constraint and the corresponding class of functions $\mathcal{F}_{\tilde{\Theta}}$. Let $\mathcal{F}_{\tilde{\Theta}}$ have global parameter symmetries $\tilde{P}$. We define a hidden symmetry as any group $H$ where $H \leq P$ and $H \nleq \tilde{P}$.

However these hidden symmetries still have consequences on the loss landscape of the constrained model. Intuitively, we may try imagining the gradient flow. In particular, assuming nicely chosen parameterization [31], once we reach a fixed point subspace we cannot leave it. For instance, once two neurons share the same weights, they would both receive the same gradient updates and would always share the same weights. But the constrained parameter space $\tilde{\Theta}$ can intersect a fixed point subspace corresponding to a hidden symmetry giving what we call a **hidden fixed point space**

$\tilde{\Theta}_H$. But we can never leave $\tilde{\Theta}_H$ through gradient descent. In particular, critical points of the loss landscape $L|_{\tilde{\theta}_H}$ would be critical points of $L|_{\tilde{\Theta}}$.

**Lemma 3.2** (Hidden symmetry induced critical points). *Let $\mathcal{F}_{\tilde{\Theta}} \subset \mathcal{F}_{\Theta}$ be a class of constrained networks. Let $L : \Theta \to \mathbb{R}$ be a loss associated to these networks. Let $H$ be a hidden symmetry with orthogonal action and let $\Theta_H = \{\theta \in \Theta : h\theta = \theta \; \forall h \in H\}$ and $\tilde{\Theta}_H = \Theta_H \cap \tilde{\Theta}$.*

*If $(\Theta_H^{\perp} \cap \tilde{\Theta}) + (\Theta_H \cap \tilde{\Theta}) = \tilde{\Theta}$, then critical points of $L|_{\tilde{\Theta}_H}$ are also critical points of $L|_{\tilde{\Theta}}$.*

The lemma requires an orthogonality condition $(\Theta_H^{\perp} \cap \tilde{\Theta}) + (\Theta_H \cap \tilde{\Theta}) = \tilde{\Theta}$. This condition holds in the case of equivariant networks when we reduce a transitive p-rep to a smaller transitive p-rep and is a relatively mild condition.

While not explored in this work, we suspect Lemma 3.2 can be helpful for understanding scaling behavior of equivariant networks using loss landscape geometry arguments similar to [46].

If the hidden fixed point subspace $\tilde{\Theta}_H$ is one dimension lower than the constrained subspace, we have an even stronger statement. In this case, $\tilde{\Theta}_H$ in some sense splits $\tilde{\Theta}$ into two halves. We cannot traverse between the two halves via gradient descent because once we reach $\tilde{\Theta}_H$ we must get stuck. Generically, we expect the global minima to be in one of these halves so if we are initialized in the "bad" half, then we would never reach the global minima. This is depicted in Figure 3. Hence, we expect there to be an additional minimum on the other side.

To formally state the theorem, we require the following conditions and discuss their satisfiability.

**Conditions:**

C1 $(\Theta_H^{\perp} \cap \tilde{\Theta}) + (\Theta_H \cap \tilde{\Theta}) = \tilde{\Theta}$
   This is used to invoke Lemma 3.2. This condition can hold in the case of equivariant networks when neuron permutation symmetries can reduce a transitive p-rep to a smaller transitive p-rep.

C2 $\dim(\tilde{\Theta}_H) = \dim(\tilde{\Theta}) - 1$
   This is a strong condition. We expect this condition can sometimes hold in the first layer but will generally fail if the previous layer is wide.

C3 There exists some $C$ such that for any direction $\hat{r} \in \tilde{\Theta}$ we have $\hat{r} \cdot \nabla L(C\hat{r}) > 0$
   This condition will be satisfied if we include regularization. This is mainly used to draw a region with a minimum in its interior.

C4 The minimum of $L|_{\tilde{\Theta}_H}$ is nondegenerate in $L|_{\tilde{\Theta}}$
   This is primarily used to ensure we can fully characterize saddle points and minima via the Hessian. When we have degeneracy (singular Hessian matrix), it may result from some additional symmetry which we can exploit to preserve the theorem.

**Theorem 3.3** (Hidden symmetry induced minima). *Let $\mathcal{F}_{\tilde{\Theta}} \subset \mathcal{F}_{\Theta}$ be a class of constrained networks. Let $L : \Theta \to \mathbb{R}$ be a loss associated to these networks. Let $H$ be a hidden symmetry with orthogonal action and let $\Theta_H = \{\theta \in \Theta : h\theta = \theta \; \forall h \in H\}$ and $\tilde{\Theta}_H = \Theta_H \cap \tilde{\Theta}$.*

*Suppose there is a minima $\theta_1$ of $L|_{\tilde{\Theta}}$ such that $\theta_1 \notin \tilde{\Theta}_H$. If the conditions above hold, there must exist a distinct minimum $\theta_2 \neq \theta_1$ of $L|_{\tilde{\Theta}}$.*

We would like to emphasize that the parameter space being split into two halves by a fixed point subspace is not considered a problem. This is because the two halves of parameter space are related by parameter symmetry and represent the same set of possible functions. So the extra minima could just be another global minima reached through parameter symmetry. However, for the constrained space, the two halves can arise from a **hidden** parameter symmetry. In this case, the two halves of the constrained space are not necessarily related by some other global parameter symmetry and the additional minima could truly be a spurious one. In Section 4 we show this can indeed be the case for equivariant networks.

### 3.2 Hidden fixed point spaces of equivariant networks

We now characterize hidden fixed point spaces of networks built using p-reps. In particular, we consider the parameter symmetries corresponding to permuting neurons. In fact, we provide a slightly

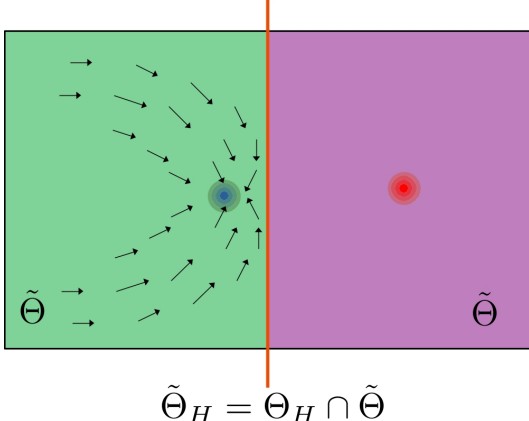

$$\tilde{\Theta}_H = \Theta_H \cap \tilde{\Theta}$$

Figure 3: Constrained subspace split into two halves by intersection with a fixed point subspace of the unconstrained model. We call the subspace separating the constrained space a hidden fixed point subspace. Gradient flows from the two halves cannot reach each other implying existence of separate minima.

more general characterization of not just the hidden fixed point spaces but the hidden "reducible" ones based on the concept of "irreducible" neurons from [46].

First, we show that for a given transitive block, any hidden fixed point subspace corresponds to a smaller transitive p-rep. We expect this scenario could negatively impact optimization.

**Theorem 3.4.** *Let a $m$-neuron point be a transitive permutation representation. Suppose the weights satisfy the corresponding equivariance constraints. If this $m$-neuron point is reducible, then either we can eliminate these neurons or we can replace the transitive permutation representation with a $n$-neuron transitive permutation representation where $n < m$ and $n|m$.*

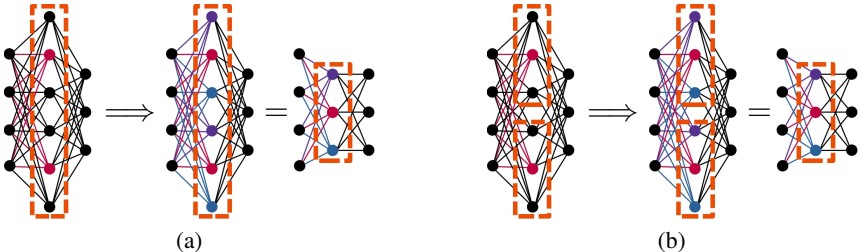

Figure 4: (a) If two neurons within a block of a transitive p-rep share input weights, then multiple other pairs of neurons must also share the same input weights. The p-rep can be reduced into a smaller one. (b) Suppose two blocks of neurons each individually are irreducible neurons and transitive p-reps. Then if two neurons share input weights, then input weights for all other neurons are shared. The two transitive p-reps can be compressed into one transitive p-rep.

Next, we show that two transitive p-reps which share neurons must collapse to effectively one transitive p-rep. It turns out this same subspace is also a fixed point subspace corresponding to permuting the entire two blocks. Hence, we do not expect it to negatively impact optimization.

**Theorem 3.5.** *Let there be transitive p-reps defined on distinct blocks of $m$-neurons and $m'$-neurons in layer $\ell$, each of which is irreducible. Suppose the $m + m'$-neuron point combining these neurons is reducible. Then they must both be the same permutation representation and we can replace the $m + m'$-neuron block with a $m$-neuron point.*

# 4 Experiments

For our experiments, we consider a teacher-student setup so we can guarantee perfect loss is achievable. We use 2-layer networks with trainable weights only between the input and hidden layer. These are of the form

$$f(x; \theta) = \sum_i \sigma \left( \sum_j \theta_{ij} x_j \right)$$

where $\sigma$ is our nonlinearity. For the experiments here, we use `ReLU`. Results with other activations can be found in Appendix E. All of our experiments can be run on a laptop.[1]

We assume data for each of our input neurons is draw independently at random from a unit normal distribution. We use the teacher networks to label the corresponding output values. For such a setup, it turns out it is possible to analytically calculate the exact loss landscape for certain activation functions [44, 48]. For our loss landscape visualization, we use this exact loss landscape.

## 4.1 Permutation equivariance

In this section, we consider permutation symmetry $S_n$ and our example uses $S_3$. In Appendix E, we also consider spherical signals used in $O(3)$ and $SO(3)$ equivariant networks.

We consider 2 types of p-reps. First, we have the trivial representation $\pi_0$ which is 1-dimensional. We then have the usual action of $S_n$ which permutes $n$ orthogonal basis vectors, which gives a $n$-dimensional representation $\pi_n$. In our code, we also have support for the regular representation $\pi_{\text{reg}}$ which is $|S_n| = n!$-dimensional and is found in group convolution [12].

Between $\pi_0$ and $\pi_n$, it is not hard to check the only possible equivariant linear map is of form $\theta \mathbf{1}$ where $\mathbf{1}$ is a $n \times 1$ matrix consisting of only 1's. Between $\pi_1$ and $\pi_1$, possible equivariant maps are of the form $\theta_1 I + \theta_2(\mathbf{1}\mathbf{1}^T - I)$ where $I$ is the identity matrix [54]. We can think of $\theta_1$ and $\theta_2$ as parameterizing diagonal and off diagonal terms respectively.

## 4.2 Loss landscape visualization

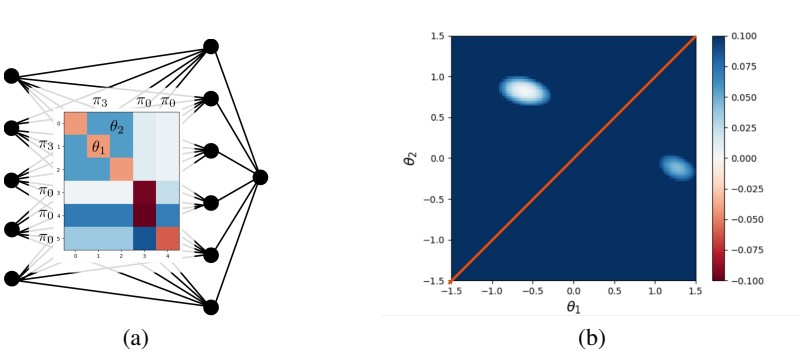

(a)                                                     (b)

Figure 5: (a) Weights of the teacher network. (b) Loss of the student network as we vary the diagonal and off diagonal weights of the $\pi_3 \to \pi_3$ map. Other weights of the student network are set equal to that of the teacher network. Red line corresponds to the fixed point subspace where $\theta_1 = \theta_2$.

For a $\pi_n$ block in the hidden layer, we note there is a hidden symmetry corresponding to permuting the corresponding $n$ neurons. The resulting hidden fixed point subspace happens exactly the weights to these neurons are all the same. If we have a single $\pi_n$ block in the input, then the linear map $\theta_1 I + \theta_2(\mathbf{1}\mathbf{1}^T - I)$ is in the hidden fixed point subspace when $\theta_1 = \theta_2$. In particular, the degrees of freedom is reduced exactly by 1 so Theorem 3.3 applies and we expect there to be an additional induced minima. Note one can check that additional $\pi_0$ representations in the input or hidden layer do not affect this.

---

[1]Code for running the experiments is available at github.com/atomicarchitects/tale-of-two-symmetries

Hence, we choose $S_3$ and a 5-dimensional input consisting of one $\pi_3$ rep and 2 trivial $\pi_0$ reps. In the hidden layer we choose one $\pi_0$ rep and 3 trivial $\pi_0$ reps. We randomly generate weights for the teacher network. We then set all other weights of the student network to equal that of the teacher network and only vary the diagonal and off diagonal weights $\theta_1$ and $\theta_2$ controlling the mapping from $\pi_3$ to $\pi_3$. Figure 5 shows the resulting loss landscape for a random seed. We clearly see two distinct minima in the loss landscape. In Appendix E, we initialize teacher weights using different random seeds and see similar results.

### 4.3 Training, bad initializations, and parameterization

For each value of $\theta_1, \theta_2$, we trained the student network for 100 steps at a learning rate of $10^{-1}$ using gradient descent on the exact loss. This is usually sufficient for the student network to converge. The resulting final loss is shown in Figure 6a. Note that indeed we see two regions, one which converges to a global minima and one which does not. However, the boundary is not a straight line. It turns out this is because gradient descent behavior depends on parameterization and in particular, on the relative scaling of the parameters [31]. By rescaling our parameterization to $\sqrt{n-1}\theta_1 I + \theta_2(\mathbf{1}\mathbf{1}^T - I)$ and then performing training, we obtain Figure 6b. Here, the boundary between the two regions is indeed a straight line and corresponds to the hidden fixed point space. We emphasize, however, that no matter how we rescale our parameters, there will always be a boundary. In Appendix E, we generate teacher weights using different random seeds and see similar results.

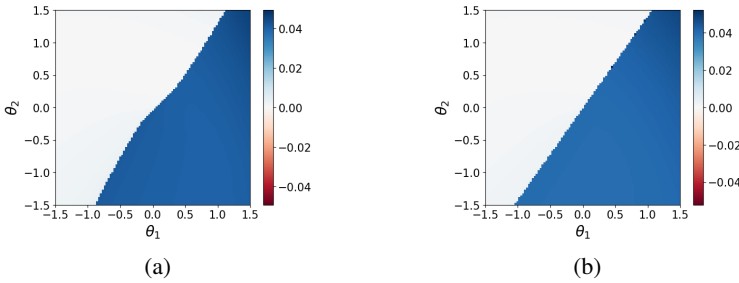

(a)                  (b)

Figure 6: Final loss achieved by student network after training for 100 steps. (a) Here we parameterize the diagonal and off diagonal components directly. The boundary is curved in this case. (b) Here we rescale the diagonal component. We have a straight line as boundary in this case which corresponds to the hidden fixed point subspace.

### 4.4 Relaxation

Using the same setup as before, we randomly choose teacher network weights and choose a bad initialization for the student network. We train the student network until it converges to the bad minima. Then, we relax the equivariance constraints completely and continue training until the network converges. The weights of the teacher network, bad minima found by the constrained network, and good minima found by the relaxed network are shown in Figure 7.

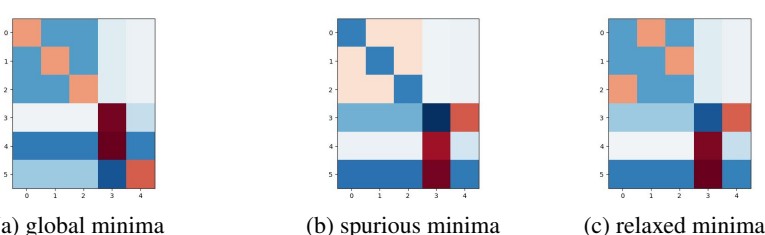

(a) global minima         (b) spurious minima         (c) relaxed minima

Figure 7: (a) Weights of the teacher network. (b) Spurious minima reached by equivariant student network after training. (c) Global minima reached by unconstrained network initialized at the spurious minima.

Note that the final weights found by the relaxed network are simply permuted with respect to the teacher weights. However, importantly this corresponds to a different group representation in the hidden layer. In order to traverse between these different choices, we must break equivariance. This process is tracked in Figure 8. We initially have a negative Hessian eigenvalue, indicating that in the relaxed network the spurious minima becomes a saddle point. As we traverse between the two different equivariant subspaces, we see a bump in equivariance error. However, our loss still decreases. Finally, we arrive at a global minima in a different equivariant subspace and we return to no equivariance error. Note that the Hessian becomes positive definite indicating that we have indeed reached a minima not a saddle.

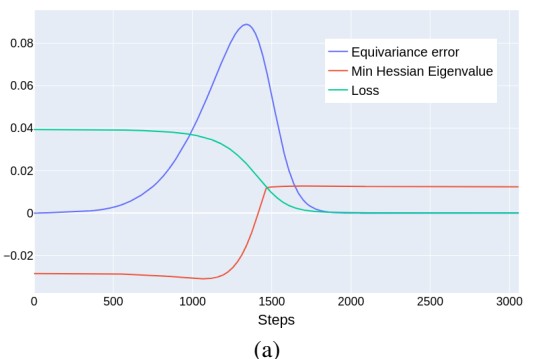

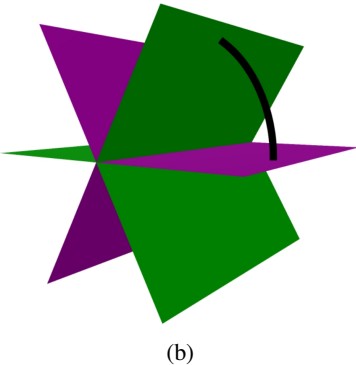

(a)                                                                                    (b)

Figure 8: Training of unconstrained network initialized at bad minima. We break out of the original equivariant subspace into a symmetrically related equivariant subspace. In particular, the "bad" half of the original constrained subspace is closer to the "good" half of the symmetrically related equivariant subspace. (a) Depicts values of various quantities during training of the relaxed model. (b) Projection of the parameters onto a 3-dimensional subspace. The planes correspond to various symmetrically related equivariant subspaces color coded by good and bad halves containing the good or bad minimum respectively.

## 5    Conclusion

In this work, we presented a general framework for investigating loss landscapes of constrained networks. In particular, we proved that the parameter symmetries of the unconstrained networks leads to hidden fixed point spaces which can create critical points and even spurious minima in the constrained networks. We then apply this to equivariant networks built with permutation representations, which can be viewed as a subspace of unconstrained MLPs. We characterize the hidden fixed point subspaces of these networks. Finally, we experimentally demonstrate the existence of these spurious minima using a teacher-student setup. Further, we empirically find that removing the equivariance constraints can sometimes convert the spurious minima to a saddle allowing the network to escape. Interestingly, the final relaxed weights are a permuted version of the teacher weights and corresponds to a different choice of group representation in the hidden layer and hence a different equivariant subspace.

There are still a number of questions to be addressed in future works. First, our results require viewing a given model as a subspace of a larger model class. Finding larger model classes which give useful insights is often challenging. In particular, it is not obvious to us how to do so for equivariant models using tensor product operations. Next, Lemma 3.2 and Theorem 3.3 only look at fixed point subspaces. However, these subspaces often have additional local parameter symmetries and we expect taking these into account can give a stronger, more broadly applicable statement. In addition, we do not concretely understand when relaxation converts bad symmetry induced minima into saddle points. Finally, we believe that exploring relaxation techniques which change our choice of group representation could prove to be immensely helpful and be a fruitful area for future studies.

## Acknowledgments and Disclosure of Funding

We acknowledge the support of the National Science Foundation under Cooperative Agreement PHY-2019786 (The NSF AI Institute for Artificial Intelligence and Fundamental Interactions). YuQing Xie was also supported by the National Science Foundation Graduate Research Fellowship under Grant No. DGE-1745302.

We would also like to thank our helpful discussions with Robin Walters, Elyssa Hofgard, Hannah Lawrence, Vasco Portilheiro, and Yuxuan Chen.

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

# A   Notation

For convenience, we provide a table of some of the commonly used symbols in our paper and their typical meaning.

Table 1: Commonly used symbols

| | |
|---|---|
| $G$ | Group our network is equivariant under |
| $X$ | Typically used to represent the space of our inputs |
| $Y$ | Typically used to represent the space of our outputs |
| $\rho$ | A group representation $\rho : G \to \mathrm{GL}(V)$ |
| $x$ | Input |
| $y$ | Output |
| $\Theta$ | Weight space of unconstrained networks |
| $f_\theta$ | Function $f_\theta : X \to Y$ parameterized by $\theta \in \Theta$ |
| $P$ | Group of parameter symmetries |
| $\tilde{\Theta}$ | Linear subspace of parameters corresponding to enforcing some constraint |
| $\tilde{P}$ | Parameter symmetries of constrained class of networks, in particular we must have $p\tilde{\Theta} \in \tilde{\Theta}$ for all $p \in \tilde{P}$ |
| $H$ | Hidden parameter symmetry group, $H \leq P$ and $H \not\leq \tilde{P}$ which arises because our choice of constraint enforcement breaks the unconstrained parameter symmetries |
| $\Theta_H$ | Fixed point subspace, we have $h\theta_H = \theta_H$ for all $\theta_H \in \Theta_H$ and $g \in H$ |
| $\tilde{\Theta}_H$ | Hidden fixed point subspace, equal to $\Theta_H \cap \tilde{\Theta}_H$ |
| $I$ | Identity matrix |
| $\mathbf{1}$ | Constant matrix of 1s |
| $\pi$ | Permutation representation. |

# B   Other related works

Previously the optimization dynamics of equivariant, non-equivariant, and non-equivariant augmented networks were studied in [38]. This work showed that the sets of critical points of an equivariant network remain critical points when we expand to an unconstrained space if we also include data augmentation. In particular, under full augmentation, the chosen equivariant subspace is invariant under gradient flow. Our work is complementary in that we show how critical points can arise in the equivariant subspace due to interactions with parameter space symmetries. In addition, our experiments show a network leaving an equivariant subspace. This is consistent with [38] since we trained using SGD where our individual batches are not fully augmented. This means we can have gradient updates which leave the equivariant subspace.

Another very interesting series of works studies families of minima in 2-layer teacher-student ReLU setups [3, 4]. This is a similar setup to that used in our toy experiments with a model of the form

$$f(x; W_1, W_2) = W_2 \sigma(W_1 x)$$

where $W_1, W_2$ are trainable weights and $\sigma$ is a pointwise nonlinearity chosen to be ReLU. In our setup, we fix $W_2$ to be a constant matrix consisting only of 1's. In [3, 4], they choose a teacher network such that $W_1 = I$ and $W_2$ consists of all 1's. The corresponding critical points and Hessian spectrum for the loss of the student network can be analyzed by exploiting symmetries of the teacher network from which they provide a characterization of various families of minima. It turns out this analysis has intimate connections with our main theorems and the results of [38]. We briefly provide a sketch of this connection below.

In particular [4] looks at $W_1$ isotropic under $\Delta S_m \times \Delta S_{n-m}$: consisting of simultaneously permuting the first $m$ rows and columns, or simultaneously permuting the last $n = m$ rows and columns. It turns out such $W_1$ correspond exactly to those satisfying equivariance constraints for $G = S_m \times S_{n-m}$ where we choose $\pi_n \times \pi_0$ and $\pi_0 \times \pi_{n-m}$ representations for the first $m$ and next $n - m$ neurons in the hidden layer. The minima families in [4] correspond exactly to those implied by our Theorem 3.3 if constrained to these symmetric $W_1$; specifically weights which have blocks where diagonal values

are less than the off diagonal ones. Because the input distribution is unit Gaussian and hence also symmetric under $G$, the results of [38] implies that even if we relax the constraints on $W_1$, the induced minima implied by our Theorem 3.3 remain critical points. Importantly however, this does not show they are minima in the unconstrained space. Only through the more detailed analysis of [4] do we know that these are minima for wide enough inputs.

## C  Group theory background

Here, we present a brief overview of concepts from group theory. We refer to standard textbooks for a more comprehensive treatment [5, 17, 18, 32]. We begin by defining what a group is.

**Definition C.1** (Group). Let $G$ be a nonempty set equipped with a binary operator $\cdot : G \times G \to G$. This is a group if the following group axioms are satsfied

1. Associativity: For all $a, b, c \in G$, we have $(a \cdot b) \cdot c = a \cdot (b \cdot c)$

2. Identity element: There is an element $e \in G$ such that for all $g \in G$ we have $e \cdot g = g \cdot e = g$

3. Inverse element: For all $g \in G$, there is an inverse $g^{-1} \in G$ such that $g \cdot g^{-1} = g^{-1} \cdot g = e$ for identity $e$.

Examples of groups include the group of permutations, the group rotation matrices with matrix multiplication as the group operation, and the group of integers under addition. One very important group is the group of automorphisms on a vector space. This group is denoted $GL(V)$ and we can think of it as the group of invertible matrices.

We care about using groups to describe symmetries. The group elements abstractly represent the symmetry operations and the effect of these operations is what we call a group action.

**Definition C.2** (Group action). Let $G$ be a group and $\Omega$ a set. A group action is a function $\alpha : G \times \Omega \to \Omega$ such that $\alpha(e, x) = x$ and $\alpha(g, \alpha(h, x)) = \alpha(gh, x)$ for all $g, h \in G$ and $x \in \Omega$.

One may want to relate two groups to each other. In particular, we would want a mapping which preserves the group structure. Such a mapping is a homomorphism.

**Definition C.3** (Group homomorphism and isomorphism). Let $G$ and $H$ be groups. A group homomorphism is a function $f : G \to H$ such that $f(u \cdot v) = f(u) \cdot f(v)$ for all $u, v \in G$. A group homomorphism is an isomorphism if $f$ is a bijection.

Usually, we consider data which lives in some vector space. Further, linear actions on vector spaces are represented by matrices. Hence, it is particular useful to relate arbitrary groups to groups consisting of matrices. Such a homomorphism together with the vector space the matrices act on is a group representation.

**Definition C.4** (Group representation). Let $G$ be a group and $V$ a vector space over a field $F$. A group representation is a homomorphism $\rho : G \to GL(V)$ taking elements of $G$ to automorphisms of $V$.

Given any representation, there are often orthogonal subspaces which do not interact with each other. If this is the case, we can break our representation down into smaller pieces by restricting to these subspaces. It is useful to consider the representations which cannot be broken down. These are known as the irreducible representations (irreps) and often form the building blocks of more complex representations.

**Definition C.5** (Irreducible representation). Let $G$ be a group, $V$ a vector space, and $\rho : G \to GL(V)$ a representation. A representation is irreducible if there is no nontrivial proper subspace $W \subset V$ such that $\rho|_W$ is a representation of $G$ over space $W$.

There has been considerable work on characterizing the irreps of various groups and many equivariant neural network designs use this knowledge.

Another special type of group representation is a permutation representation. It turns out only these types of representations are compatible with arbitrary pointwise nonlinearities [39].

**Definition C.6** (Permutation representation). Let $G$ be a group. Any group homomorphism $\pi : G \to \mathrm{Sym}(n)$ is an abstract permutation representation. Let $\rho : \mathrm{Sym}(n) \to \mathrm{GL}(V_n)$ be the representation

mapping to corresponding permutation matrices. Then $\rho \circ \pi : G \rightarrow \mathrm{GL}(V_n)$ is a permutation representation. We will abbreviate permutation representation as p-rep.

The intuition is that for a p-rep, there is always some choice of basis $b_i$ such that $\rho(g)b_i = b_{\pi(i)}$. In other words, group actions just permute the basis. The trivial representation is an example since there is only one basis and it remains invariant under group action. The regular representation commonly used in group convolutions is another example since we can associate each basis vector to a group element [12].

Just like how irreps are building blocks of arbitrary representations, we are interested in the building blocks of p-reps. Such p-reps are the transitive p-reps. For such p-reps, each pair of basis vectors can be transformed into each other through a group action.

**Definition C.7** (Transitive permutation representation). Let $G$ be a group and $\rho : G \rightarrow \mathrm{GL}(V_n)$ be a permutation representation. We say the permutation representation is transitive if there is a basis $b_1, \ldots, b_n$ of $V_n$ such that the action of $G$ on the set $\{b_1, \ldots, b_n\}$ is transitive.

It turns out transitive p-reps can also be characterized and it is much simpler than for irreps. To do so, we must introduce some additional concepts.

First, we introduce subgroups. A natural question is whether a subset of group elements themselves form a group under the same group operation. Such a subset is a called a subgroup.

**Definition C.8** (Subgroup). Let $G$ be a group and $S \subseteq G$. If $S$ together with the group operation of $G \cdot$ satisfy the group axioms, then $S$ is a subgroup of $G$ which we denote as $S \leq G$.

One particular feature of a subgroup is that we can use them to decompose our group into disjoint chunks called cosets.

**Definition C.9** (Cosets). Let $G$ be a group and $S$ a subgroup. The left cosets are sets obtained by multiplying $S$ with some fixed element of $G$ on the left. That is, the left cosets are for all $g \in G$

$$gS = \{gs : s \in S\}.$$

We denote the set of left cosets as $G/S$. The right cosets are defined similarly except we multiply with a fixed element of $G$ on the right. That is, the right cosets are for all $g \in G$

$$Sg = \{sg : s \in S\}.$$

We denote the set of right cosets as $G\backslash S$.

Next, it is useful to define what we mean by symmetry of an object. These are all group elements which leave the object unchanged and is called the stabilizer.

**Definition C.10** (Stabilizer). Let $G$ be a group, $\Omega$ some set with an action of $G$ defined on it, and $u \in \Omega$. The stabilizer of $u$ is all elements of $G$ which leave $u$ invariant. That is

$$\mathrm{Stab}_G(u) = \{g : gu = u, g \in G\}.$$

One can check that the stabilizer is indeed a subgroup. Closely related to the stabilizer is the orbit. This is all the values we get when we act with our group on some object.

**Definition C.11** (Orbit). Let $G$ be a group, $\Omega$ some set with an action of $G$ defined on it, and $u \in \Omega$. The orbit of $u$ is the set of all values obtained when we act with all elements of $G$ on it. That is,

$$\mathrm{Orb}_G(u) = \{gu : g \in G\} = Gu.$$

It turns out one can show that the stabilizer of elements in the orbit are related. This relation turns out to be conjugation which we define below.

**Definition C.12** (Conjugate subgroups). Let $G$ be a group and $S$ and $S'$ be subgroups. We say $S$ and $S'$ are conjugate if there is some $g \in G$ such that

$$gSg^{-1} = S.$$

Conjugacy of subgroups defines an equivalence relation.

**Definition C.13** (Conjugate classes of subgroups). Let $G$ be a group and $S$ a subgroup. Denote the conjugate subgroups of $S$ as the set

$$\mathrm{Cl}_G(S) = \{gSg^{-1} | g \in G\}.$$

The conjugate classes of subgroups are defined as the different sets of conjugate subgroups.

*Remark* C.14. The conjugate classes of subgroups for finite subgroups of $O(3)$ correspond to the point groups. This is the reason why certain groups such as $D_3$ and $C_{3v}$ are named differently despite being isomorphic. No $D_3$ subgroup can be transformed into a $C_{3v}$ subgroup via conjugation by some element of $O(3)$.

Now we can characterize the transitive permutation representations.

**Lemma 2.6.** *Let $G$ be a group. There is a canonical bijection between the transitive permutation representations of $G$ and the classes of conjugate subgroups.*

*Proof.* First, given any conjugate class of subgroups $C$, we can always pick a subgroup $S \in C$. Then consider the cosets $G/S$ and pick coset representatives $\{g_1, \ldots, g_n\}$. We can construct a basis $b_1, \ldots, b_n$ and define the action of $g \in G$ on $b_i$ as $gb_i = b_j$ where $j$ is such that $g_j = gg_iS$. Hence, we can map each conjugate class of subgroups to a transitive permutation representation.

Let $\rho : G \to \mathrm{GL}(V_n)$ be a transitive permutation representation. Suppose $b_1, \ldots, b_n$ is a basis of $V_n$ where $\rho$ consists of permutation matrices. Then consider the mapping $\rho$ to $\mathrm{Cl}_G(\mathrm{Stab}_G(b_1))$. Note it is easy to check that this inverts our previous construction of a t-rep from a conjugate class.

Now consider some different transitive permutation representation $\rho' : G \to \mathrm{GL}(V_n)$ where $b_1', \ldots, b_n'$ is a basis where $\rho'$ consists of permutation matrices. Further, suppose $\mathrm{Cl}_G(\mathrm{Stab}_G(b_1')) = \mathrm{Cl}_G(\mathrm{Stab}_G(b_1))$.

Then there must be some $g \in G$ such that $g\mathrm{Stab}_G(b_1)g^{-1} = \mathrm{Stab}_G(b_1')$. Then define a mapping given by $\pi(g'b_1') = g'gb_1$ for all $g' \in G$. Note that $\mathrm{Stab}_G(gb_1) = g\mathrm{Stab}_G(b_1)g^{-1} = \mathrm{Stab}_G(b_1')$ which means this map is consistent. Note we can express $\pi$ as a change of basis matrix $B$. But then

$$B^{-1}\rho(g'')B(g'b_1) = B\rho(g'')g'gb_1 = B^{-1}(g''g')gb_1 = g''g'b_1' = \rho'(g'')g'b_1'.$$

Hence, $\rho$ and $\rho'$ are equivalent representations.

Lastly, we state Schur's lemma which is a very useful tool for characterizing equivariant linear maps.

**Theorem C.15** (Schur's Lemma). *Let $G$ be a group and $V, W$ be 2 vector spaces equipped with irreps $\rho_V, \rho_W$.*

1. *If $\rho_V, \rho_W$ are not isomorphic, then there are no nonzero $G$-equivariant maps $V \to W$*

2. *If $\rho_V, \rho_W$ are finite dimensional over an algebraically closed field and $\rho_V = \rho_W$, then the only notrivial $G$-equivariant linear maps $V \to W$ are scalar multiples of the identity.*

## C.1 Infinite permutation representations

Permutation representations are typically defined in the finite case. Here we show how this definition can be extended for infinite dimensional representations.

First, recall that the abstract permutation representation was defined as a homomorphism $\pi : G \to \mathrm{Sym}(n)$. We can view $\mathrm{Sym}(n)$ as $\mathrm{Aut}(X)$ for some set of cardinality $|X| = n$. We can instead replace the finite $X$ with some arbitrary (possibly infinite set).

**Definition C.16.** Let $X$ be some set. Define a homomorphism $\pi : G \to \mathrm{Aut}(X)$ as a generalized abstract permutation representation.

We can then view any function $f : X \to \mathbb{F}$ where $\mathbb{F}$ is some field as an infinite dimensional vector space. In particular, we can consider some $\mathbb{F}^X$ with some canonical basis $c_x$ so that any $v \in \mathbb{F}^X$ can be written as

$$v = \sum_{x \in X} f(x)c_x.$$

Using the generalized abstract p-rep $\pi$, we can define an action on this space

**Definition C.17.** Let $G$ be a group, $X$ be some space, $\mathbb{F}$ be a field, and $\pi : G \to X$ a generalized abstract p-rep. Then we can define a map $\rho : G \to \mathrm{GL}(\mathbb{F}^X)$ by

$$\rho(g)v = \rho(g) \sum_{x \in X} f(x)c_x = \sum_{x \in X} f(x)c_{\pi(g)x} = \sum_{x \in X} f(\pi^{-1}(g)x)c_x$$

where $v = \sum_{x \in X} f(x)c_x \in \mathbb{F}^X$ is any vector. This is what we call a generalized permutation representation.

An example of a generalized p-rep is $S^2$ signals. If we embed $S^2$ as a unit sphere in $\mathbb{R}^3$, then the standard rotation action $R$ on $\mathbb{R}^3$ gives a natural generalized abstract p-rep. In particular, we can simply define $\pi : SO(3) \to \mathrm{Aut}(S^2)$ by

$$\pi(R)\mathbf{x} = R\mathbf{x}.$$

We simply just rotate the sphere.

Hence by Definition C.17, for any spherical signal $f : S^2 \to \mathbb{R}$, the action $\rho(g)[f](x) = f(R^{-1}x)$ defines a generalized p-rep on the infinite dimensional space of spherical signals. $\square$

# D   Proofs

**Lemma 3.2** (Hidden symmetry induced critical points). *Let $\mathcal{F}_{\tilde{\Theta}} \subset \mathcal{F}_\Theta$ be a class of constrained networks. Let $L : \Theta \to \mathbb{R}$ be a loss associated to these networks. Let $H$ be a hidden symmetry with orthogonal action and let $\Theta_H = \{\theta \in \Theta : h\theta = \theta \; \forall h \in H\}$ and $\tilde{\Theta}_H = \Theta_H \cap \tilde{\Theta}$.*

*If $(\Theta_H^\perp \cap \tilde{\Theta}) + (\Theta_H \cap \tilde{\Theta}) = \tilde{\Theta}$, then critical points of $L|_{\tilde{\Theta}_H}$ are also critical points of $L|_{\tilde{\Theta}}$.*

*Proof.* We first note that any linear invertible action on $\Theta$ induces an action on the tangent space. If $M \in \mathrm{GL}(\Theta)$ is an invertible matrix, then this induces an action on a function $f : \Theta \to \mathbb{R}$ given by $f(\theta) \to f'(\theta) = f(M^{-1}\theta)$. By chain rule, we find that

$$(\nabla f'(\theta))_i = \partial_i f'(\theta) = \partial_k f(M^{-1}\theta) \cdot \partial_{\theta_i} M_{kj}^{-1}\theta_j = (\nabla f(M^{-1}\theta))_k M_{kj}^{-1}$$

where we used Einstein summation conventions. Viewing $\nabla$ as a row vector we get

$$\nabla f'(\theta) = (M^{-1})^T \nabla f(M^{-1}\theta)$$

so there is an additional action of $(M^{-1})^T$ in the tangent space.

Because $L$ is symmetric under $H$, we have

$$\nabla L(\theta) = (h^{-1})^T \nabla(L(h\theta)) = (h^{-1})^T \nabla(L(\theta))$$

Since we assume $H$ has orthogonal action, then $h^{-1} = h^T$ so,

$$\nabla L(\theta) = (h^{-1})^T \nabla(L(\theta)) = h\nabla L(\theta)$$

By definition of $\Theta_H$, this means we must have $\nabla L(\theta) \in \Theta_H$ for any $\theta \in \Theta_H$.

Suppose $\theta^*$ is a critical point of $L_{\tilde{\Theta}_H}$. In particular, for $\theta^*$ we have $\nabla L(\theta^*) \in \Theta_H$. For any direction $t \in \tilde{\Theta}$, we can uniquely decompose

$$t = t_H + t_H^\perp$$

for $t_H \in \Theta_H$ and $t_H^\perp \in \Theta_H^\perp$. But since $\tilde{\Theta} = (\Theta_H^\perp \cap \tilde{\Theta}) + (\Theta_H \cap \tilde{\Theta})$, we must have $t_H \in (\Theta_H \cap \tilde{\Theta})$ and $t_H^\perp \in (\Theta_H^\perp \cap \tilde{\Theta})$.

Therefore,

$$\begin{aligned} t \cdot \nabla L(\theta^*) &= (t_H + t_H^\perp)\nabla L(\theta^*) \\ &= t_H \nabla L(\theta^*) \\ &= 0 \end{aligned}$$

since we showed $\nabla L(\theta^*) \in \Theta_H$ and we assumed $\theta^*$ is a critical point of $\tilde{\Theta}_H$. $\square$

**Lemma D.1** (Minima on boundary). *Let $U$ be an open subset of $\mathbb{R}^n$. Suppose $f : \bar{U} \to \mathbb{R}$ is a $C^1$ smooth function. Consider a point $x \in \partial U$ on the boundary which is locally $C^1$ smooth so that we can define an outward pointing unit normal $\hat{n}(x)$. Then if $\hat{n}(x) \nabla f(x) > 0$ then $x$ cannot be the global minima of $f$.*

*Proof.* Let $\delta = \hat{n}(x) \nabla f(x)$. There exists some $\epsilon$ such that for all $u$ where $|u - x| < \epsilon$ we have $|f(u) - f(x) - (u - x) \cdot \nabla f(x)| < \frac{\epsilon \delta}{2}$. In particular, suppose we choose $u = x - \frac{\epsilon}{2} \hat{n}(x)$. Then

$$
\begin{aligned}
f(u) &< f(x) + (u - x) \cdot \nabla f(x) + \frac{\epsilon \delta}{2} \\
&= f(x) - \frac{\epsilon}{2} \hat{n}(x) \cdot \nabla f(x) + \frac{\epsilon \delta}{2} \\
&\leq f(x) - \frac{\epsilon}{2} \delta + \frac{\epsilon \delta}{2} \\
&= f(x).
\end{aligned}
$$

Therefore $x$ cannot be the global minima. $\qquad\square$

**Lemma D.2** (Existence of minima in functions on balls). *Let $\bar{B}^n$ be a closed $n$-dimensional ball. Suppose $f : \bar{B}^n \to \mathbb{R}$ is a $C^1$ smooth function. Suppose for any point $x$ on the boundary $\partial \bar{B}^n$ we have $\hat{n}(x) \cdot \nabla f(x) > 0$ where $\hat{n}$ is the normal vector out of the boundary. Then $f$ has an absolute minima in the interior $B^n$.*

*Proof.* First, because $f$ is a continuous function over a compact space, it has an absolute minima by the extreme value theorem. Further, since the boundary is $C^1$ smooth and $\hat{n}(x) \cdot \nabla f(x) > 0$, Lemma D.1 tells us there cannot be minima on the boundary. Hence we conclude the global minima of $f$ lies in the interior $B^n$.

$\qquad\square$

**Theorem 3.3** (Hidden symmetry induced minima). *Let $\mathcal{F}_{\tilde{\Theta}} \subset \mathcal{F}_\Theta$ be a class of constrained networks. Let $L : \Theta \to \mathbb{R}$ be a loss associated to these networks. Let $H$ be a hidden symmetry with orthogonal action and let $\Theta_H = \{\theta \in \Theta : h\theta = \theta \,\forall h \in H\}$ and $\tilde{\Theta}_H = \Theta_H \cap \tilde{\Theta}$.*

*Suppose $(\Theta_H^\perp \cap \tilde{\Theta}) + (\Theta_H \cap \tilde{\Theta}) = \tilde{\Theta}$ and $\dim(\tilde{\Theta}_H) = \dim(\tilde{\Theta}) - 1$. Suppose there is a minima $\theta_1$ of $L|_{\tilde{\Theta}}$ such that $\theta_1 \notin \tilde{\Theta}_H$. Further suppose that there exists some $C$ such that for any direction $\hat{r} \in \tilde{\Theta}$ we have $\hat{r} \cdot \nabla L(C\hat{r}) > 0$. Then if Hessians of all critical points are nondegenerate, there must exist a distinct minima $\theta_2 \neq \theta_1$ of $L|_{\tilde{\Theta}}$.*

*Proof.* First, consider the closed ball $\bar{B}_C(0) = \{\theta \in \tilde{\Theta}_H : |\theta| \leq C\}$. By assumption, on the boundary we have $\hat{r} \cdot \nabla L(C\hat{r}) > 0$ so Lemma D.1 tells us $L$ has a minima $\theta^*$ in the interior $B_C(0)$. So $\theta^*$ must be a critical point and in fact must be a local minima of $L|_{\tilde{\Theta}_H}$. But by Lemma 3.2, $\theta^*$ is also a critical point of $L|_{\tilde{\Theta}}$.

If $\theta^*$ is a local minima of $L|_{\tilde{\Theta}}$, then we have an additional minima distinct from $\theta_1$ since $\theta_1 \notin \tilde{\Theta}_H$ but $\theta^* \in \tilde{\Theta}$.

So suppose $\theta^*$ is not a local minima of $L|_{\tilde{\Theta}}$. Let $\hat{n} \in \tilde{\Theta}$ be a unit vector normal to $\tilde{\Theta}_H$. Then we can consider two hemispheres given by

$$
\bar{D}_1 = \{x \in \tilde{\Theta} : x \cdot \hat{n} \leq 0, \ |x| \leq C\}
$$

$$
\bar{D}_2 = \{x \in \tilde{\Theta} : x \cdot \hat{n} \geq 0, \ |x| \leq C\}
$$

We claim neither of these sets can have minima on their boundaries. Without loss of generality consider $D_1$. Its boundary consists of $S_1 = \{x \in \tilde{\Theta} : x \cdot \hat{n} < 0, \ |x| = C\}$ and $\bar{B}_C(0)$. Along $S_1$, by assumption $\hat{n}(x) \cdot \nabla L(x) > 0$ so Lemma D.2 tells us the global minima of $\bar{D}_1$ cannot be on $S_1$. On $\bar{B}_C(0)$, we know the minimum value is $L(\theta^*)$. However, we assumed $\theta^*$ is not a minima so it must be a saddle. Since we assumed nondegenerate Hessians at critical points, there must be some unit vector $\hat{d}$ such that $\hat{d}^T H(\theta^*) \hat{d} < 0$. Note that since $\theta^*$ is a minima of $L_{\tilde{\Theta}_H}$, $|\hat{d} \cdot \hat{n}| > 0$. Without loss of generality suppose $\hat{d} \cdot \hat{n} < 0$.

Then for any $\delta$ there exists some $\epsilon$ such that for all $\theta \in \tilde{\Theta}$, if $|\theta - \theta^*| < \epsilon$ then $|L(\theta) - L(\theta^*) - \frac{1}{2}(\theta - \theta^*)^T H(\theta^*)(\theta - \theta^*)| < |\theta - \theta^*|^2 \delta$. Pick $\delta = \frac{1}{4}|\hat{d}^T H(\theta^*)\hat{d}|$ and $\theta = \theta^* + \epsilon\hat{d}/2$. Since $\hat{d} \cdot \hat{n} < 0$, we see $\theta \cdot \hat{n} < 0$ so $\theta \in \bar{D}_1$. But we see that

$$
\begin{aligned}
L(\theta) <& L(\theta^*) + \frac{1}{2}\frac{\epsilon\hat{d}^T}{2}H(\theta^*)\frac{\epsilon\hat{d}}{2} + \frac{\epsilon^2}{4}\delta \\
=& L(\theta^*) + \frac{1}{2}\epsilon^2(-\delta) + \frac{\epsilon^2}{4}\delta \\
=& L(\theta^*) - \frac{1}{4}\epsilon^2\delta \\
<& L(\theta^*).
\end{aligned}
$$

Therefore there is a point in the interior of $\bar{D}_1$ with value smaller than $L(\theta^*)$. Hence the minima also cannot be on $\bar{B}_C(0)$. Hence $L|_{\bar{D}_1}$ must have its global minima in the interior. Similarly $L|_{\bar{D}_2}$ must as well so there are 2 distinct local minima. $\qquad\square$

**Theorem 3.4.** *Let a $m$-neuron point be a transitive permutation representation. Suppose the weights satisfy the corresponding equivariance constraints. If this $m$-neuron point is reducible, then either we can eliminate these neurons or we can replace the transitive permutation representation with a $n$-neuron transitive permutation representation where $n < m$ and $n|m$.*

*Proof.* Let $W$ be the input weights with dimensions $m \times d_{\text{in}}$ and $U$ be the output weights with dimensions $d_{\text{out}} \times m$. Since the hidden neurons transform as a permutation representation, for any $g \in G$ denote by $\pi_g(i)$ as the node which $g$ transforms node $i$ into. Denote by $\rho, \rho', \rho''$ the representations of $G$ on the current $m$-neurons ($\mathbb{R}^m$), previous layer ($\mathbb{R}^{d_{\text{in}}}$), and next layer ($\mathbb{R}^{d_{\text{out}}}$) respectively.

There are 2 cases for reducibility. Either we have a neuron with 0 output weight or a pair of neurons which share input weights.

The first case is that there is some $k$ where $U_{ik} = 0$ for all $i \in [d_{\text{out}}]$. In this case we find that by equivariance constraints for any $g \in G$ we have

$$
U_{i\pi_g(k)} = (U\rho(g))_{ik} = (\rho''(g)U)_{ik} = 0.
$$

Therefore all output weights are 0 and we can eliminate this transitive block of neurons.

The second case is that there exist distinct $k, k'$ such that $W_{ki} = W_{k'i}$ for all $i \in \{1, 2, \ldots, d_{\text{in}}\}$. Suppose $h \in G$ is such that $\pi_h(k') = k$. Then by equivariance constraints we must have

$$
(\rho(h)W)_{ki} = W_{\pi_{h^{-1}}(k)i} = W_{k'i} = (W\rho'(h))_{ki} = W_{kj}\rho'_{ji}(h)
$$

where we use Einstein summation notation. But since $W_{k'i} = W_{ki}$ we have

$$
W_{ki} = W_{kj}\rho'_{ji}(h) \tag{1}
$$

for all such $h$. Further, let $S_k$ be the stabilizer of $k$ under $\pi$. That we have $\pi_{s^{-1}}(k) = k$ for all $s \in S_k$ and $i \in \{1, 2, \ldots, d_{\text{in}}\}$. Then by equivariance constraints we also have

$$
(\rho(s)W)_{ki} = W_{ki} = W_{kj}\rho'_{ji}(s). \tag{2}
$$

Now, consider a group $A$ generated by elements of $S_k$ and $h$. Then any $a \in A$ can be written as

$$
a = g_1 g_2 \ldots g_p
$$

for some finite $p$ where each $g_i$ is either in $S_k$ or is $h$. We then find that

$$
\begin{aligned}
W_{kj}\rho'_{ji}(a) =& W_{kj_1}\rho'_{j_1 j_2}(g_1) \ldots \rho'_{j_p i}(g_p) \\
=& \left(W_{kj_1}\rho'_{j_1 j_2}(g_1)\right) \ldots \rho'_{j_p i}(g_p) \\
=& W_{kj_2}\rho'_{j_2 j_3}(g_2) \ldots \rho'_{j_p i}(g_p) \\
=& \ldots \\
=& W_{ki}
\end{aligned}
$$

where we repeatedly apply (1) and (2). However, by equivariance we must also have

$$W_{ki} = W_{kj}\rho'_{ji}(a) = (\rho(a)W)_{ki} = W_{\pi_{a^{-1}}(k)i}$$

for all $i \in \{1, 2, \ldots, d_{\text{in}}\}$ and any $a \in A$.

Next, consider some coset $gA \in G/A$. We then find for any $b = ga \in gA$ we have

$$
\begin{aligned}
W_{\pi_b(k)i} &= (\rho(b^{-1})W)_{ki} = W_{kj}\rho'_{ji}(b^{-1}) \\
&= W_{kj}\rho'_{ji}(a^{-1}g^{-1}) \\
&= W_{kj_1}\rho'_{j_1 j_2}(a^{-1})\rho'_{j_2 i}(g^{-1}) \\
&= W_{kj_2}\rho'_{j_2 i}(g^{-1}).
\end{aligned}
$$

Hence, we see that for any $g$ from the same coset, the incoming weights $W_{\pi_g(k)i}$ are the same. Since our permutation representation is transitive, all neurons are reached by appropriate choice of $g \in G$. Hence, we can reduce our $m$-neuron point to a $n = |G/A|$-neuron point.

Finally, we must show that with this reduction we still satisfy equivariance constraints. To do so, let us define our new weights and new representation. First, pick some coset representatives $\{g_1 = e, g_2, \ldots, g_n\}$. Next, we define new ingoing weights

$$W'_{pq} = W_{\pi_{g_p}(k)q}.$$

and outgoing weights

$$U'_{qp} = \sum_{g \in g_p A} U_{q\pi_g(k)}.$$

Let $\tilde{\rho} : G \to \mathrm{GL}(\mathbb{R}^{|G/A|})$ be permutation matrices defined as

$$
\tilde{\rho}_{ij}(g) = \begin{cases} 1 & gg_j \in g_i A \\ 0 & \text{otherwise} \end{cases}
$$

One can check that these matrices indeed form a permutation representation. Further, denote by $\tilde{\pi}_g : [|G/A|] \to [|G/A|]$ to be the corresponding permutation of indices for $\tilde{\rho}(g)$.

We then check for any $g \in G$ that

$$
\begin{aligned}
(W'\rho'(g))_{pq} &= W'_{pi}\rho'_{iq}(g) = W_{\pi_{g_p}(k)i}\rho'_{iq}(g) \\
&= \rho_{kj}(g_p^{-1})W_{ji}\rho'_{iq}(g) = \rho_{kj}(g_p^{-1})\rho_{ji}(g)W_{iq} \\
&= \rho_{ki}(g_p^{-1}g)W_{iq} = W_{\pi_{g^{-1}g_p}(k)q}
\end{aligned}
$$

Now let $g_j$ be such that $g^{-1}g_p \in g_j A$. Then we see

$$
\begin{aligned}
(W'\rho'(g))_{pq} &= W_{\pi_{g_j}(k)q} = W'_{jq} \\
&= W'_{\tilde{\pi}_{g^{-1}}(p)q} = \tilde{\rho}_{pi}(g)W'_{iq} \\
&= (\tilde{\rho}(g)W')_{pq}.
\end{aligned}
$$

So $W'$ satisfies the equivariance constraints.

For $U'$ we check that

$$
\begin{aligned}
(\rho''(g)U')_{qp} &= \rho''_{qi}(g)U'_{ip} \\
&= \sum_{g' \in g_p A} \rho''_{qi}(g)U_{i\pi_{g'}(k)} \\
&= \sum_{g' \in g_p A} U_{qi}\rho_{i\pi_{g'}(k)}(g) \\
&= \sum_{g' \in g_p A} U_{qi}\rho_{ij}(g)\rho_{jk}(g') \\
&= \sum_{g' \in g_p A} U_{qi}\rho_{ik}(gg')
\end{aligned}
$$

Let $g_j$ be such that $gg_p \in g_j A$. Then for any $g' \in g_p A$, we have $g' = g_p a$ for some $a \in A$ so $gg' = gg_p a = g_j a' a$ since we know $gg_p = g_j a'$ for some $a'$. So then $gg' \in g_j A$. Hence we see that

$$
\begin{aligned}
(\rho''(g)U')_{qp} &= \sum_{g'' \in g_j A} U_{qi}\rho_{ik}(g'') \\
&= U'_{qj} = U'_{q\tilde{\pi}_g(p)} \\
&= U'_{qi}\tilde{\rho}_{ip}(g) \\
&= (U'\tilde{\rho}(g))_{qp}.
\end{aligned}
$$

So the new output weights also satisfy equivariance constraints. $\square$

**Theorem 3.5.** *Let a $m$-neuron point and a $m'$-neuron point be distinct transitive permutation representations each of which are irreducible. Suppose the $m + m'$-neuron point combining these neurons is reducible. Then they must both be the same permutation representation and we can replace the $m + m'$-neuron point with a $m$-neuron point.*

*Proof.* Let use denote the incoming weights to the $m$-neuron and $m'$-neuron points as $W$ and $W'$ respectively. Suppose $k, k'$ are such that

$$
W_{ki} = W'_{k'i}
$$

for all $i \in [d_{\text{in}}]$.

First, we show that both the $m$ and $m'$ neuron points are the same permutation representation. Let

$$
S = \text{Stab}_G(k) \qquad S' = \text{Stab}_G(k').
$$

Suppose $S \neq S'$. Then either there is some $s \in S$ where $s \notin S'$ or there is some $s' \in S$ but $s' \notin S$. Without loss of generality suppose we have the former. Then by equivariance, we have

$$
\begin{aligned}
W_{kj} &= (\rho(s^{-1})W)_{kj} = (W\bar{\rho}(s^{-1}))_{kj} \\
&= W_{ki}\bar{\rho}_{ij}(s^{-1}) = W'_{k'i}\bar{\rho}_{ij}(s^{-1}) \\
&= (W'\bar{\rho}(s^{-1}))_{k'j} = (\rho'(s^{-1})W')_{k'j} \\
&= W'_{\pi'_s(k')j}.
\end{aligned}
$$

But since $s \notin S'$, $\pi'_s(k') \neq k'$. But this would mean $m'$-neuron point is reducible, a contradiction. Hence we must have $S = S'$. Because both neuron points are transitive, this implies they must be the same.

To conclude, we note that equivariance constraints uniquely define the remaining rows of $W, W'$ which means there is a bijection between rows of $W, W'$ where all corresponding rows are equivalent. Hence, we can combine all these neurons can create a single $m$-neuron point instead. $\square$

# E  Additional experiments

## E.1  Permutation symmetry

Using the same setup as in Section 4, we tried different random seeds for initializing the teacher weights.

### E.1.1  Loss landscape

In Figure 9, we have the loss landscape as we vary the diagonal and off diagonal components of the $\pi_3 \to \pi_3$ map for various random seeds. There seem to be two distinct minima in most of the plots. For Figures 9b, 9i, 9j, the distinct minima is not very apparent. By directly checking the weights, we were able to confirm that the diagonal and off diagonal components of the teacher weights are very close which likely led to this phenomenon. In addition, for Figures 9e and 9g, the minima are out of range of our parameter sweep. Interestingly, we note that the global and spurious minima always appear to be roughly equidistant from the diagonal line corresponding to the hidden fixed point subspace.

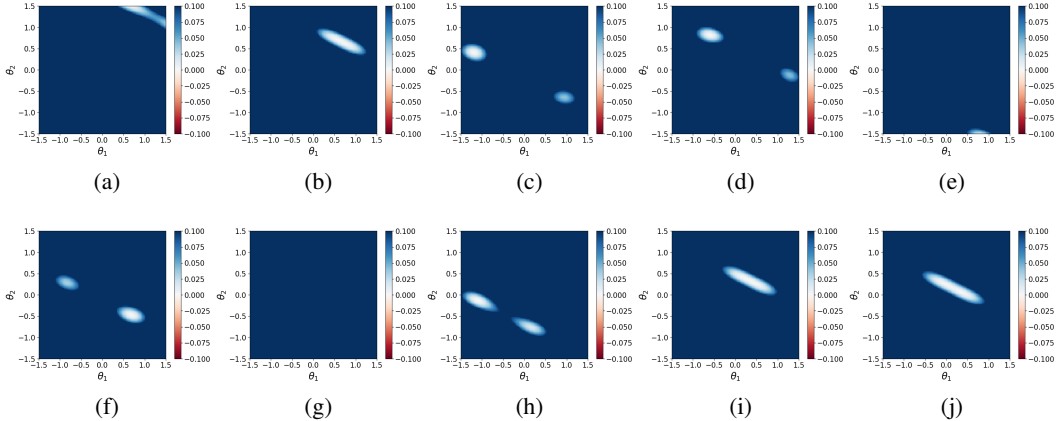

Figure 9: Loss landscapes for various random seeds. Note there seems to be two distinct minima in almost all cases.

### E.1.2 Constrained training

Same as before, we initialize student networks to have the same weights as the teacher network except for the diagonal and off diagonal components of the $\pi_3 \to \pi_3$ map. We then train for 100 steps with gradient descent and record the final loss. We do this both for a direct parameterization of diagonal and off diagonal values (Figure 10 and one with rescaled diagonal values (Figure 11). As before, we see clear boundaries in most of the cases and in Figure 10 we generally see a nonlinear boundary while in Figure 11 we generally see a linear one which corresponds to the hidden fixed point space. However, for (b) and (i) there seems to be a different boundary which is much more prominent. It is unclear to us exactly where this boundary comes from. We suspect that there may be additional spurious minima not explained by our symmetry argument. Such minima are known to be common in these 2-layer ReLU setups [45].

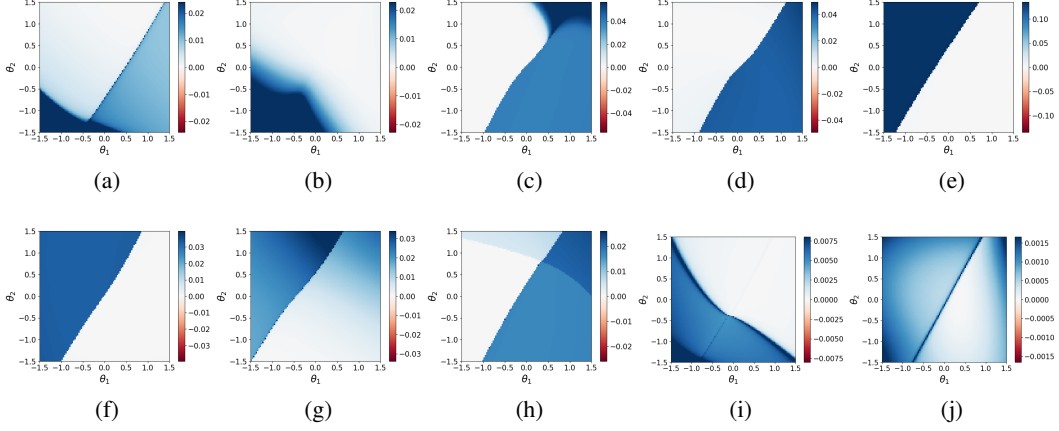

Figure 10: Loss after training student network for 100 steps. We directly parameterize the $\pi_3 \to \pi_3$ map as $\theta_1 I + \theta_2 (\mathbf{1}\mathbf{1}^T - I)$.

### E.2 Different activation

Finally, we also tried replacing the `ReLU` activation with an `erf` activation. This was chosen because there is an analytical loss for a teacher-student setup for iid unit Gaussian inputs [44]. The results are shown in Figure 12. We also tend to see additional minima as expected from theory.

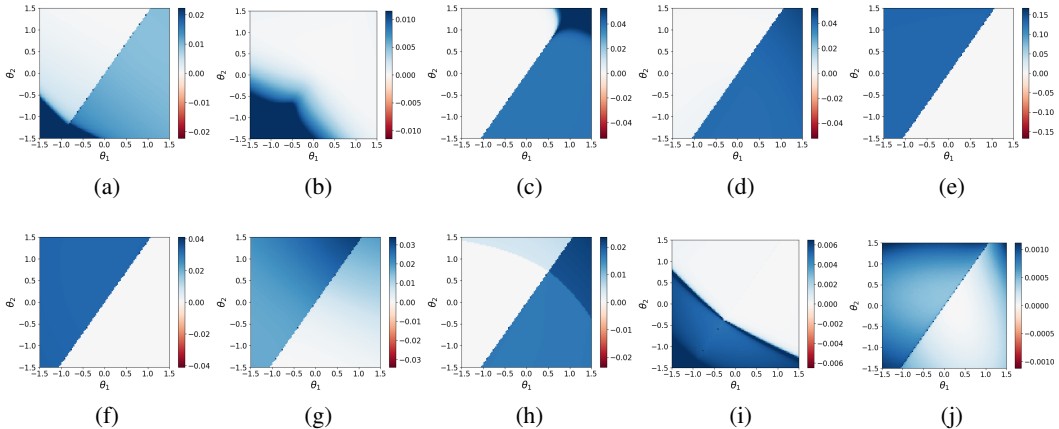

Figure 11: Loss after training student network for 100 steps. We rescale the $\pi_3 \to \pi_3$ map as $\sqrt{n-1}\theta_1 I + \theta_2(\mathbf{1}\mathbf{1}^T - I)$.

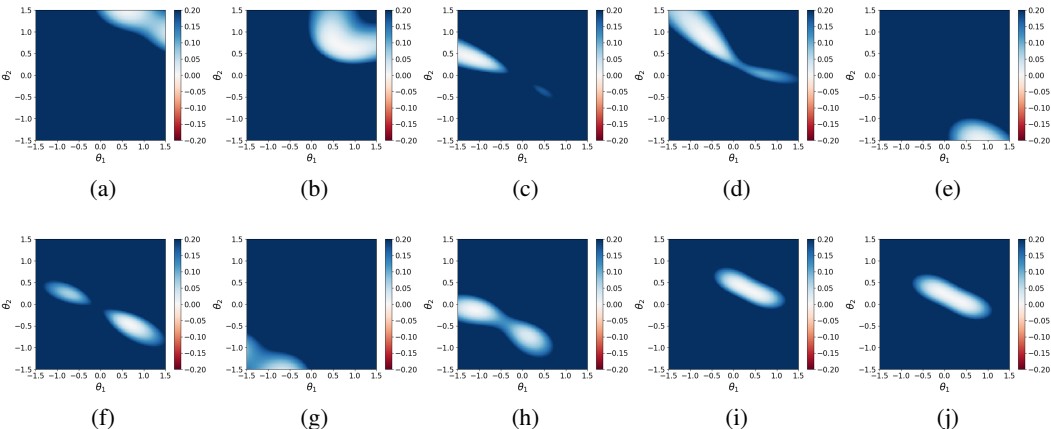

Figure 12: Loss landscapes for various random seeds. Note there seems to be two distinct minima in almost all cases.

### E.3 $S^2$ signals

It turns out spherical signals are also permutation representations. This is a result of $\mathrm{SO}(3)/\mathrm{SO}(2) \cong S^2$. Spherical signals are used in $S^2$ activations first proposed in [13]. In many equivariant archi-tectures, spherical signals are parameterized in the Fourier basis using spherical harmonics [26, 47]. Next, we note that any automorphism of $S^2$ in principle gives a hidden symmetry and the corre-sponding fixed point subspace is a constant signal. This corresponds to only a scalar $\ell = 0$ spherical harmonic. Hence, if our inputs only consists of scalar $\ell = 0$ spherical harmonic and a single nonscalar spherical harmonic, Theorem 3.3 applies and we expect multiple minima.

To test, this, we construct a simple model. First, we pick some cutoff $L_{\max}$ for maximum degree of spherical harmonic used. We then pick some representation for our input by specifying the relevant $\mathrm{SO}(3)$ irreps. We construct an equivariant linear map from our input representation to $0 \oplus 1 \oplus \ldots \oplus L_{\max}$ used to represent the $S^2$ signal. We then apply our nonlinearity on the sphere and then extract the harmonics of the final signal. Finally we dot product final harmonics with itself and sum to obtain our output. We use the e3nn library to build this model [26].

### E.3.1 Single scalar and nonscalar irrep

We first consider the case where our input rep is of the form $0 \oplus L$. This is the case where Theorem 3.3 applies. For such a network, by Schur's lemma there are only two degrees of freedom in the equivariant linear layer. We map the $0$ and $L$ irreps to the $\ell = 0$ and $\ell = L$ harmonics of the spherical signal. Mathematically, we have the signal

$$f(\hat{r}) = \theta_0 x^{(0)} + \theta_1 \sum_m x_m^{(L)} Y_m^L(\hat{r})$$

where $x^{(0)}$ is the scalar input and $x_m^{(L)}$ is the $L$ irrep input and $\theta_0, \theta_1$ are the weights.

The hidden fixed point space corresponding to automorphisms of the sphere is precisely constant signals on the sphere. This happens when $\theta_1 = 0$. Hence by Theorem 3.3, we expect there to be two minima corresponding to positive and negative values of $\theta_1$. We randomly choose values of $\theta_0, \theta_1$ for the teacher network then sweep over $\theta_0, \theta_1$ and compute the loss of the student network. The resulting loss landscape is shown in Figure 13.

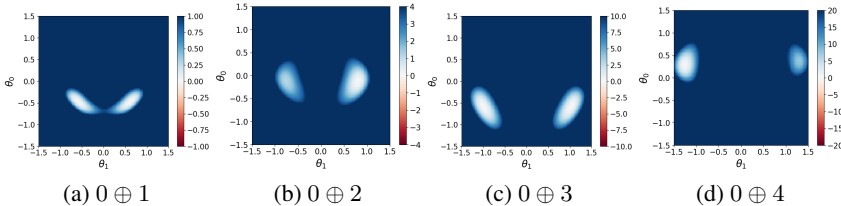

| (a) $0 \oplus 1$ | (b) $0 \oplus 2$ | (c) $0 \oplus 3$ | (d) $0 \oplus 4$ |

Figure 13: Loss landscapes for various random seeds and different $L$. Again, note the existence of two minima. However, the landscape appears symmetric for odd $L$ but not even $L$.

Note that the loss landscape appears symmetric for odd values of $L$. It turns out that this is because the inversion map on $S^2$ gives a simple sign flip on odd spherical harmonic values. Hence this is a parameter symmetry of such a model and our previous hidden fixed point space is in fact just a fixed point space for odd $L$. So for odd $L$ we expect the two minima to be the same. However, for even $L$, the $\theta_1 = 0$ line is truly a hidden fixed point subspace and we expect different minima.

### E.3.2 Other input reps

We also tried testing other combinations of input reps for which Theorem 3.3 does not apply. In particular, we consider inputs of type $L_1 + L_2$ such that for $x_m^{(L_1)}$ and $x^{(L_2)}$ input we obtain a signal

$$f(\hat{r}) = \theta_1 \sum_m x_m^{(L_1)} Y_m^{L_1}(\hat{r}) + \theta_2 \sum_m x_m^{(L_2)} Y_m^{L_2}(\hat{r})$$

where $\theta_1, \theta_2$ are the weights. We again randomly initialize these weights for the teacher network and sweep over $\theta_1, \theta_2$ for the student network and compute the loss. Interestingly, we still often observe multiple minima reminiscent of the patterns expected from our other experiments.

## F  Loss landscape and different optimizers

We provide a brief discussion of how our loss landscape characterization affects our understanding of optimization. In general, because Theorem 3.3 implies existence of multiple minima, we expect potential issues where gradient based optimizers converge to the wrong minimum. In addition, we note that for certain "nice" parameterization, gradients in the hidden fixed point space stay in that space. Hence gradient descent and SGD would never cross this space. If we model weight decay with L2-regularization, we see that the regularization term does not affect the symmetry of the loss landscape so this barrier remains. However, for different parameterizations or equivalently, methods assigning different learning rates to different parameters [31], the hidden fixed point space could pose less of an issue.

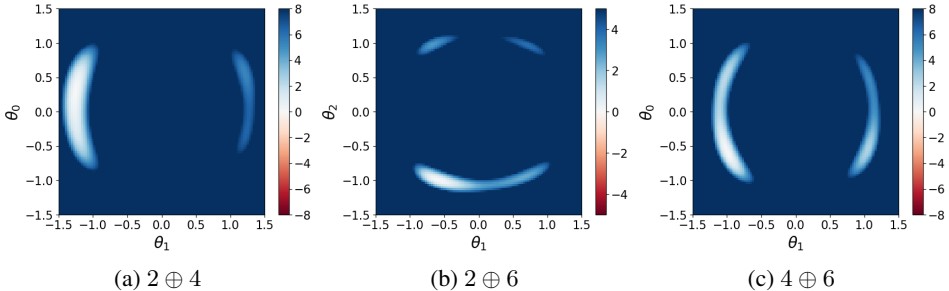

(a) $2 \oplus 4$          (b) $2 \oplus 6$          (c) $4 \oplus 6$

Figure 14: Loss landscapes for various random seeds and different input types $L_1 \oplus L_2$. Even though Theorem 3.3 does not apply, we still often observe multiple minima.

Lastly, Lemma 3.2 implies existence of critical points in a much more general setting. While these may be saddles instead of minima, saddles are still considered a problem for many optimization methods [16]. Further, a number of works have shown saddle-to-saddle dynamics under various settings [1, 28, 42]. Hence we believe Lemma 3.2 may have important consequences to be explored in future work.

