# OpenReview forum: "A Tale of Two Symmetries: Exploring the Loss Landscape of Equivariant Models"
_NeurIPS.cc/2025/Conference — NeurIPS 2025 poster_

### Official Review · Reviewer_AZWo · 2025-06-10

**Clarity:** 1
**Significance:** 3
**Originality:** 3
**Rating:** 5
**Confidence:** 3

**Summary:**

The authors study the effect of symmetries in the parameter space on the loss landscape of neural networks. The authors argue that any symmetry (that can be what they call a 'hidden symmetry') will induce possibly spurious local minima. More specifically, they prove that if the subspace $\Theta_H$ of invariant weights under the group divides the parameter space in two half-spaces, any minimum must be accompanied by a distinct one. They showcase that this result goes beyond the relatively clear fact that if $\theta$ is a minimum and $h$ is a global parameter symmetry, then $h\theta$ is also one via numerical experiments.

**Questions:**

1.  If $\tilde{\Theta}\subseteq \Theta$, mustn't the symmetry group $P$ for $\Theta$ (as defined in Definition 2.7) be a subgroup of the symmetry group $\tilde{P}$ of $\tilde{\Theta}$ (if applying $p \in P$ leaves every function $f_\theta$ unaffected, that is in particular true for $f_\theta, \theta\in \tilde{\Theta}$? But if $P \leq \tilde{P}$, there are no $H$ with $H\leq P$ but $H\not\leq P$.
  In the paragraph just before Definition 2.7, it seems like the authors use a different definition for global parameter symmetries for the set $\tilde{P}$: A parameter symmetry here seems to be defined as the $h$ so that $h\tilde{P}\subseteq \tilde{P}$. Can the authors clarify?

2. Probably cause by the above, I do not understand what the hidden symmetries are in the numerical experiments. In the text, the authors say that these are the 'permutations within the block'. In which ways are these permutations defined? It seems to me that the only ways the weights in the layer depicted in Figure 4 can be permuted within the block while keeping the function the same is by permuting the rows and columns simultaneously. However, for this action, the matrix stays the same for all values of $\theta_1$ and $\theta_2$, so that this does not seem to be a 'hidden symmetry'. It is in particular not clear to me why the subspace $\theta_1=\theta_2$ is invariant in this case (which seems to be crucial in the proof of Theorem 3.3). Does not figure 5(a) indicate that it is not?

3. The formulations of Theorem 3.4 and 3.5 are different in the appendix and main paper. In the main paper, 'neuron blocks' are permutation representations, where as 'neuron points' are permutation representations. Which one is correct? It is for me slightly unclear in which sense a single point in parameter space (which I assume is what is meant by a neuron point) can be a representation of the permutation group. On the other hand, reducability seems to be defined for neuron points, making the meaning of reducability of neuron blocks in the formulation in the main paper hard to understand. Can the authors clarify?
Along the same lines, when reparametrizing $(\theta_1,\theta_2) \mapsto (\sqrt{n-1}\theta_1,\theta_2)$, the invariant subspace should change ($\theta_1=\theta_2$ is not invariant under that reparametrization), but according to Figure 5(b), it does not. Can the authors clarify?

4. Can the authors comment between the relation between their work and [1] (see above)?

5. In the appendix, the authors claim that spherical signals are permutation representations, but then seem to argue that $\mathrm{SO}(3)$ acts on functions on the sphere. There seems to be a mismatch here, can the authors explain?

**Ethical Concerns:**

["NO or VERY MINOR ethics concerns only"]

**Final Justification:**

The author's rebuttal made the meaning of the paper much clearer to me, and I can know appreciate it a lot better. With this added understanding, I now definitely believe that it should be accepted.

**Limitations:**

The authors could make it clearer that the condition $\mathrm{dim}(\widetilde{\Theta}_H)= \mathrm{dim}(\widetilde{\Theta})-1$ is somewhat restrictive. In the cases the authors consider in their figures and experiments, $\dim(\Theta)$ is often 2, so that $\mathrm{dim}(\widetilde{\Theta}_H)= \mathrm{dim}(\widetilde{\Theta})-1$ is plausible. It is unclear what the situation is in more involved cases.

**Paper Formatting Concerns:**

The paper consistently writes 'minima' (which is plural) instead of 'minimum' (which is singular). This should be corrected.

**Quality:**

2

**Strengths And Weaknesses:**

The main  very interesting observations. The fact that symmetries seem to induce spurious local minimum is a new and potentially impactful result. The proof of Theorem 3.3 and the lemmata leading up to it are easy to read and correct. The proofs of Theorem 3.4 and 3.5 also seem correct, although there are some questions about their formulation, see questions below.

The results (and idea of viewing equivariant nets as restrictions of nets in larger parameter spaces) seem to have tight relations with [1], which is about the relation between training a non-constrained neural network on group-symmetric (augmented) data and the 'layerwise restriction strategy' to constructing equivariant networks. While that work does not speak about hidden symmetries (and in particular do not arrive at a result as 3.3), many of the geometrical arguments seem similar. In particular, [1] also prove a version of Lemma 3.2 in their setting.

Where the article lacks is in its readability. The authors start by motivating their work by the debate about whether equivariance should be obtained via restriction of architectures or by relying on symmetries in the data, but then quickly transition to mainly focus on other types of symmetry (namely those obtained by transforming the weights of the network). This is not wrong, but is slightly confusing.
Along the same lines, there are some crucial concepts that could be explained more thoroughly. In particular, the definition of hidden symmetries is for me slightly unclear. See more detailed questions below.

Possibly related to the above, there are some aspects of the numerical experiments that are highly unclear to me, making me doubt their relevance to the theory part of the paper. I have included some questions about this below also.

The above outlined unclarities are significant enough for me to not be able to recommend acceptance, but I am very willing to change my score if the question marks are straightened out during rebuttal.

[1] Nordenfors, O., Ohlsson, F., & Flinth, A. (2025). Optimization dynamics of equivariant and augmented neural networks. Transactions on Machine Learning Research.

---

> ### Author Rebuttal · Authors · 2025-07-31
>
> We thank the reviewer for their detailed and insightful comments. We appreciate that the reviewer finds our results novel and potentially impactful. We address the questions as they detail the concerns raised as weaknesses. We hope this clears some of the confusion.
>
> ## Questions
> 1. This is a very good question and is the heart of how we define hidden symmetries. You are absolutely correct that for any $\tilde{\theta}\in\tilde{\Theta}$ and $p\in P$ we have $f_{p\tilde{\theta}}=f_{\tilde{\theta}}$. The part that can break is line 128 in Definition 2.7, we require $p$ to be a homeomorphism. While originally $p:\Theta\to\Theta$, when restricting to $\tilde{\Theta}$ all we know is $p\vert_{\tilde{\Theta}}:\tilde{\Theta}\to p(\tilde{\Theta})$. The range of $p(\tilde{\Theta})$ is not necessarily $\tilde{\Theta}$! So $p|_{\tilde{\Theta}}$ is not necessarily a homeomorphism! We mention that this is the issue in line 180-181. A concrete example is a permutation transforming the weights in Figure 6a and 6c. Both weights describe the same function but the weight sharing structure is different.
>
>       We do not quite understand the question about the paragraph before Definition 2.7. We only start using $\tilde{P}$ in Section 3.1. We would appreciate clarification about what the reviewer meant.
>
> 2. Thanks for asking! First we clarify what happens when permuting neurons. Suppose the $W_\ell$ are the ingoing weights to neurons in layer $\ell$ and the outgoing weights are $W_{\ell+1}$. Then if we permute the neurons, we transform $W_\ell\to PW_\ell$ and $W_{\ell+1}\to W_{\ell+1}P^T$, that is we permute the rows of the ingoing weights and columns of the outgoing weights, two **different** matrices. Perhaps the reviewer is confusing this parameter symmetry with equivariance constraints, in which we transform both rows and columns of the same matrix.
>
>      As an example, in Figure 4, $W_\ell$ are the depicted weights and $W_{\ell+1}$ is a constant matrix unaffected by any permutations. In particular, permuting the first 3 rows of $W_\ell$ is a hidden symmetry and would bring us away from the constrained subspace (see difference between Figure 6a and 6c, both would give the exact same function but 6c has a different weight sharing scheme and does not lie in the original equivariant subspace). It is only where $\theta_1=\theta_2$ that this permutation has no effect.
>
> 3. We apologize for the confusion. Your interpretation of neuron point is correct. We originally borrowed terminology from [2] which uses the term neuron points and defines irreducibility of neuron points. What we intended for a neuron block was a neuron point that satisfies the equivariance constraints corresponding to a specific choice of transitive permutation representation on a set of neurons. Perhaps the following is more clear:
>
>      > Suppose we define a transitive permutation rep on a block of $m$ neurons. Consider a $m$-neuron point which satisfies the corresponding equivariance constraints. If this $m$-neuron point is reducible, then the reduced $n$-neuron point satisfies the equivariance constraints for a corresponding transitive permutation rep on the reduced block of $n$ neurons. Further $n|m$.
>
>      Yes, you are correct that the subspace is no longer $\theta_1=\theta_2$ but $\sqrt{n-1}\theta_1=\theta_2$. In Figure 5b, this is why the boundary line does not have slope $1$ but instead slope $\sqrt{2}$.
>
> 4. Thanks for bringing up this work! In [1], the main results are how critical points of unconstrained, equivariant, and augmented networks are related, followed by a discussion of their respective stabilities. However, there is no discussion on how these critical points arise which is why this result is different from Lemma 3.2.
>
>      In our work, the focus is on describing a general mechanism (hidden symmetries and hidden fixed point spaces) which gives rise to critical points (Lemma 3.2) and can even imply the existence of additional minima (Theorem 3.3). We view the hidden symmetry mechanism in Lemma 3.2 as a generalization of obtaining critical points from regular parameter symmetries [2,3]. The hidden symmetry mechanism is then applied to equivariant MLP layers built with permutation representations (Theorems 3.4 and 3.5 + experiments).
>
>      One additional note we want to make is that [1] focuses on a particular equivariant subspace since it assumes a specific choice of representation on the hidden layers. In our work, we have the interesting empirical observation that unconstrained models can learn to change the representation in the hidden layers.
> 5. This is a great question! What we mean is that spherical signals satisfy a generalized version of permutation representation.
>
>      In a more general definition, we could consider an abstract permutation representation as homomorphism $G\to\mathrm{Aut}(X)$ where $X$ is some object (maybe a set, or maybe it has some additional structure). In the case where $X$ is a finite set, then $\mathrm{Aut}(X)\cong\mathrm{Sym}(|X|)$ and we recover the usual definition of an abstract permutation representation. In the case of spherical signals, we take $X$ to be $S^2$ and the abstract permutation representation is the action of rotations on the sphere.
>
>      To move from an abstract perm rep to a perm rep, we then construct a vector space (over field $\mathbb{R}$ for us) with elements of $X$ as the basis. Then any point in the space is of the form $\sum_{x\in X}c_x x$ where $c_x\in\mathbb{R}$ are the coefficients. But we can also interpret $c_x$ as a function $X\to\mathbb{R}$. For finite $X$, the induced representation on $\\{X\to\mathbb{R}\\}$ gives us the usual $|X|$-dimensional perm reps. For infinite $X$ such as $S^2$, we instead have an induced representation on the infinite dimensional vector space of $\\{S^2\to\mathbb{R}\\}$ which corresponds precisely to the spherical signals.
>
>      We can make this point more clear in the appendix.
>
> ## Limitations
> Thanks for this suggestion. Indeed the dimension condition is somewhat restrictive. We plan to add a discussion of how restrictive each of these conditions are.
>
> However, we would like to point out that Lemma 3.2 is very general and can be applied for any linearly constrained model regardless of width and depth. We believe the identification of these hidden symmetry induced critical points is very important even if they are not minima or imply minima (as happens for the additional criteria of Theorem 3.3). This is because saddle points are a known problem for optimization [4].
>
> We thank you again for your time in reviewing our work. Please let us know if you have any other concerns.
>
> ## References
>
> [1] Nordenfors, Oskar, Fredrik Ohlsson, and Axel Flinth. "Optimization Dynamics of Equivariant and Augmented Neural Networks." Transactions on Machine Learning Research.
>
> [2] Simsek, Berfin, François Ged, Arthur Jacot, Francesco Spadaro, Clément Hongler, Wulfram Gerstner, and Johanni Brea. "Geometry of the loss landscape in overparameterized neural networks: Symmetries and invariances." In International Conference on Machine Learning, pp. 9722-9732. PMLR, 2021.
>
> [3] Fukumizu, Kenji, and Shun-ichi Amari. "Local minima and plateaus in hierarchical structures of multilayer perceptrons." Neural networks 13, no. 3 (2000): 317-327.
>
> [4] Dauphin, Yann N., Razvan Pascanu, Caglar Gulcehre, Kyunghyun Cho, Surya Ganguli, and Yoshua Bengio. "Identifying and attacking the saddle point problem in high-dimensional non-convex optimization." Advances in neural information processing systems 27 (2014).

---

> > ### Comment · Reviewer_AZWo · 2025-08-03
> > **A clarifying answer**
> >
> > I would like to thank the authors of this paper for this detailed answer. It has done a lot for me to understand the results of the paper. I was indeed confused by the two different types of symmetries involved: The ones related to the data (which can be used to define certain interesting $\Theta$) and the ones related to transformations on the weights. If I understand it correctly, the former type of symmetry should in this work only be seen as a way to define interesting $\Theta$ (and since this is done in practice, an important realm of application of the results), and the real interest lies in how it can affected by hidden symmetries. This distinction may be made clearer in the final version of the manuscript, if the authors wish.
> >
> > In particular, I know think that I understand what a hidden symmetry is: a transformation that does not affect the function, but transforms data in the 'subspace of allowed weights' $\Theta$ to points 'not allowed by the architecture' is now much more clear to me. This straightens out my confusion of the experiments and also the formulations and proofs of the theorems. In the meaning of 'permutations within the blocks' has been made much clearer to me by the rebuttal.
> >
> > I also agree that the action of $\mathrm{SO}(3)$ on the function of the sphere can be seen as an abstract permutation representation - the confusion stemmed from the fact that the permutations are $\mathrm{S}_n$-elements in the rest of the text.
> >
> > It is nice to see that the authors acknowledge that the dimension condition is restrictive, and plan to add a discussion on this. I also agree that Lemma 3.2 is interesting in its own right.
> >
> > My increased understanding has made me better appreciate the manuscript. I now definitely think that it should be accepted, and will change my score accordingly.

---

> > > ### Author Response · Authors · 2025-08-04
> > >
> > > We thank you again for your time spent evaluating our work and we truly appreciate your helpful feedback! We are grateful that our response has clarified your understanding of our work and for your endorsement!

---

### Official Review · Reviewer_Xb7d · 2025-06-28

**Clarity:** 3
**Significance:** 3
**Originality:** 3
**Rating:** 4
**Confidence:** 3

**Summary:**

This work studies the effect of equivariant constraints of the loss landscape of neural networks. Their main result establishes that hidden fixed point space can split the loss landscape of neural networks in two. This creates two unconnected regions, hindering efficient optimization if the global optimum is in the opposite region of initialization. They apply this result to permutation equivariant networks (subset of MLPs) by characterizing their hidden fixed point space. Finally they empirically confront their result to the training of small MLPs by highlighting the existence of such spurious minima and that removing the constraint can improve the optimization.

**Questions:**

- Some notations were undefined. For-example I could not see where $V_n$ (line 95) was defined. In the same way $\Theta$ (line 128) was not defined: is it a vector space, a group, a ring? I assume it is a vector space of the trainable parameters but it should be defined. Finally, in theorem 3.3, what is the definition of dimension $dim$? Is it Haussdorf, Minkowsky dimension or the usual dimension of a subvector space? This would assume that $\Theta_H$ is a sub-vector space. Is it proved if used? Finally where do you use the asssumption on the dim in the proof? (page 17)
- Can you provide some discussion on the satisfiability of the various assumptions used in your theoretical results? For example theorem 3.3 assumes "Suppose dim(  Θ_H ) = dim( Θ) − 1" that "there exists some C such that for any direction r ∈  Θ we have r · ∇L(C r) > 0" and that "Hessians of all critical points are nondegenerate." These are three assumptions that could be potentially very strong. It would be nice to add some discussion on their satisfiability in some practical or toy cases.
- One more open question: what is the effect of over-parametrization on these spurious local minima? Can over-parametrization erase these or make these be asymptotically global minima in the limit of large width?
- Finally I suggest adding a related work section since related work was only mainly discussed in the introduction.

**Ethical Concerns:**

["NO or VERY MINOR ethics concerns only"]

**Final Justification:**

I believe the results of this work are interesting and novel.

The main weaknesses to me are:

*  the strength of the assumptions used in the statement of theorem 3.3.
*  the assumption lacking to theorem 3.3 to apply lemma 3.2. On that point, it is my understanding that although it requires an additional potentially quite strong assumption in the proof, it is a fixable issue.

Finally, even though the assumptions of theorem 3.3 clearly do not apply to deep neural networks, I believe that it is not disqualifying for experiments to not satisfy the assumptions of the proof.

**Limitations:**

The authors briefly discussed limitations in the introduction. A main limitation in my opinion is that this work focuses on permutation representation, while there is a significant body of works focusing on rotation or translation symmetries. Could the author add some discussion on how would it impact the theory? Are there major limitations in adapting the proofs?

**Paper Formatting Concerns:**

This paper has no formatting concern that I noticed.

**Quality:**

3

**Strengths And Weaknesses:**

**Strengths**

- the motivation for this work is clearly explained in sections 1, 2.
- the focus of this work on whether equivariance constraints can create spurious minima seems new and theoretically interesting.
- the paper is overall well-written, with self-contained mathematical lemmas and propositions.
- the main results (thm 3.3, 3.4, 3.5) are theoretically interesting and confronted to experiments in section 4 especially figures 5, 6, 7.

**Weaknesses**

- some notations and mathematical objects are undefined which make it hard to fully understand the assumptions used.
- Assumptions are correctly stated but there is no discussion on their practical aspect.

---

> ### Author Rebuttal · Authors · 2025-07-31
>
> We thank the reviewer for their insightful comments. We appreciate that the reviewer finds our work well motivated, novel, theoretically interesting, and well written. We hope our response addresses the weaknesses and questions raised.
>
> ## Weaknesses:
> * Thank you for letting us know and for pointing out some of this in the questions! We plan to add a section in the appendix explaining our notation. We will also double check and update the main body to make sure we define everything.
> * Thanks for raising this point! We will make sure to address how strong our assumptions are in the revised version.
>
>      In particular, we would like to highlight that Lemma 3.2 is very general and can be applied for any linearly constrained model regardless of width and depth. We believe the identification of these hidden symmetry induced critical points is very important even if they are not minima or imply minima (as happens for the additional criteria of Theorem 3.3). This is because saddle points are a known problem for optimization [1].
>
> ## Questions:
> * Here, $V_n$ is supposed to be the vector space on which we define our group representation and you are correct for $\Theta$. We mean the usual dimension of a vector space by $\mathrm{dim}$. That $\Theta_H$ is a subvector space is elementary enough that we believe it can be omitted. However, your comment did make us realize we assume a linear group action which we will add in the revised version. We have written a short proof below.
>
>      > For any $\theta_1,\theta_2\in\Theta_H$, any $a_1,a_2\in\mathbb{R}$, and any $h\in H$ we have $h(a_1\theta_1+a_2\theta_2)=a_1h\theta_1+a_2h\theta_2=a_1\theta_1+a_2\theta_2$ where we used linearity of the group action. Therefore $a_1\theta_1+a_2\theta_2\in\Theta_H$ so $\Theta_H$ is a vector space.
>
>      This is a very good question! The assumption is used in line 616 where we state the boundary consists of $S_1$ and $\bar{B}_C(0)$.
>
> * This is a very good suggestion! The dimension condition is quite strong. However the other conditions are quite weak. We would also like to highlight that the dimension condition is only necessary for Theorem 3.3. Lemma 3.2 only requires a mild geometric condition which is broadly applicable.
>
> * This is a great question! We expect increased width to mitigate the spurious minima issue. First, in some cases the dimension condition is less likely to be satisfied so the spurious minima is no longer guaranteed. Secondly, with wider layers, there is a greater chance that some components of the model are initialized in the “correct” half so the model can learn to prioritize those components.
>
>      Finally, we would like to add that Lemma 3.2 and Theorem 3.3 lays the foundation for future work which may investigate how the number of non-global minimum critical points grows as we overparameterize, similar in spirit to [2]. This could potentially give insight on differences in scaling behavior between equivariant and nonequivariant models.
>
> * Thanks for this suggestion! We will make sure to do this, if accepted, with the extra content page
>
> ## Limitations
> * We want to emphasize that the hidden symmetry mechanism in Section 3.1 is quite general and independent of equivariance. While we do focus on applying this to finite groups and permutation representations in the rest of the main body, we do also have some results for rotation symmetry in Appendix C.3. In that case, we show $S^2$-activations which use spherical signals (an analog of a permutation representation) can also suffer from the same problems identified in the paper. Note that $S^2$ activations are used in real models such as EquiformerV2.
>
>      Finally, the main difficulty for other architectures is figuring out the right way to embed it as a constrained subspace of a more general model class. For permutation representations with pointwise nonlinearities, this is easy since MLPs naturally give the more general class. For translation symmetry, we note that convolutions can also be thought of as a restriction of a MLP and we at least expect Lemma 3.2 to apply. However, finding a useful unconstrained model class for layers with tensor products is more difficult.
>
> We thank you again for your time in reviewing our work. Please let us know if you have any other concerns.
>
> ## References
>
> [1] Dauphin, Yann N., Razvan Pascanu, Caglar Gulcehre, Kyunghyun Cho, Surya Ganguli, and Yoshua Bengio. "Identifying and attacking the saddle point problem in high-dimensional non-convex optimization." Advances in neural information processing systems 27 (2014).
>
> [2] Simsek, Berfin, François Ged, Arthur Jacot, Francesco Spadaro, Clément Hongler, Wulfram Gerstner, and Johanni Brea. "Geometry of the loss landscape in overparameterized neural networks: Symmetries and invariances." In International Conference on Machine Learning, pp. 9722-9732. PMLR, 2021.

---

> ### Comment · Reviewer_Xb7d · 2025-08-04
>
> I thank the authors for their reply to my questions.
>
> I thank the authors for clarifying my questions, especially the one stating (in their global answer):
>
> "*Mild assumption. This just says if we go in any direction of parameter space far enough, the loss will keep increasing. This is reasonable as we expect very large parameter values to give divergent predictions. Further, the purpose of this assumption is to show we can draw a region with no boundary critical points, which can happen even if the assumption is false."*
>
> To clarify that, if I understand well, any regularization term can enforce this condition. Otherwise, without regularization, it does not seem trivial to me why it should be true? Can you comment on that? I believe that it's not because the weights are large that the loss is also large? If for example only two weights are very large, but whose contribution could cancel, it may not hold?
>
> Also just to clarify by "for any direction \hat{r}\in \tilde{\Theta}" you mean any unitary vector? It could be good to clarify it if yes.

---

> > ### Author Response · Authors · 2025-08-04
> >
> > We thank you again for your time spent evaluating our work and we truly appreciate your helpful feedback! We are grateful that our response has clarified most of your questions!
> >
> > Regarding the remaining question, you are absolutely correct! We were originally thinking that a random direction will lead to divergence (which is true with probability 1 in many cases) but you astutely point out that there can be constructions where the weights cancel each other. Indeed, a regularization term can enforce this condition and makes it a reasonable assumption. We have edited the general response accordingly.
> >
> > Lastly, yes, by direction we mean a unit vector and will clarify.

---

> > > ### Comment · Reviewer_Xb7d · 2025-08-08
> > >
> > > I thank the authors for their response.
> > >
> > > Based on their reply and the discussion with the other reviewers, I believe it is important that the authors clearly state the strong limitations of their assumptions and amend the statement of Theorem 3.3 to include the missing assumption from Lemma 3.2.
> > >
> > > It seems to me that these changes, while important, should not require substantial alterations to the work. Given that the results are novel and interesting I will maintain my score leaning toward acceptance, on the condition that the required revisions are made if the paper is accepted.

---

> > > > ### Author Response · Authors · 2025-08-08
> > > >
> > > > We thank you again for your time spent evaluating our work and we truly appreciate your helpful feedback! We also thank you for recognizing the proposed changes are relatively minor modifications and we will make sure to clearly discuss the limitations of each assumption if accepted.

---

### Official Review · Reviewer_jLby · 2025-07-03

**Clarity:** 4
**Significance:** 2
**Originality:** 3
**Rating:** 5
**Confidence:** 3

**Summary:**

The authors identify that global parameter symmetries in networks without symmetry constraints may become “hidden symmetries” when restricted to equivariant subspaces. This can cause gradient descent to get trapped in a spurious local minima. The phenomena is studied for shallow networks using permutation representations in a teacher-student setting, where they show that relaxing the constraint changes the spurious minima to a saddle point.

**Questions:**

- Does the identified phenomena occur for real equivariant networks? Beyond a teacher-student setting?
- Are the other loss landscape problems where your tools might be useful?
- Does weight decay and/or optimizers that normalise gradients solve this issue?

**Ethical Concerns:**

["NO or VERY MINOR ethics concerns only"]

**Final Justification:**

- The paper has significant conceptual novelty.
- I disagree with the AC that because the experiments lack certain assumptions used in the theoretical results, they are invalidated. I agree with the authors and reviewer AZWo that showing that the conceptual contribution extends to a "messier" "real-world" setting in fact makes this a potentially stronger contribution.

- I would like to strongly encourage the authors to follow the ACs recommendations, and communicate Thm 3.3 with the proper limitations (from my reading, I also assume it can be fixed).

**Quality:**

4

**Strengths And Weaknesses:**

### Strengths

- Very clearly explain a novel phenomenon where equivariant constraints for a model “interact” with parameter symmetries to induce spurious minima.
- Introduce a new tool to understand the loss landscape of constrained models, where viewing the model as a member of a larger class of unconstrained models can identify “hidden symmetries” that can e.g. cause spurious minima.
- The student-teacher experiments are well constructed and support the theoretical arguments.

### Weaknesses

- The focus is on transitive permutation representations, and it’s not clear at all how well the tools / problems will generalize to practical models (e.g., this issue is mentioned in the conclusion w.r.t. tensor products). Likewise, the Hessians of “real” GNNs are typically degenerate, so it’s not clear how relevant hidden symmetry induced minima are in “practice.” Similar complains can be made for the co-dimension assumption.

---

> ### Author Rebuttal · Authors · 2025-07-31
>
> We thank the reviewer for their positive evaluation of our work! We appreciate that the reviewer finds our explanation of the novel hidden symmetry phenomenon clear and the experiments well constructed. We hope our response addresses the weaknesses and questions raised.
>
> ## Weaknesses
> * First, while we do focus on models with permutation representations, we emphasize that the results in Section 3.1 are independent of equivariance and these results apply to any architecture we can view as a constrained version of a more general model.
>
>      Whether hidden symmetry induced minima happen in practice is a very good question. First, we would appreciate a reference for degenerate Hessians in real GNNs, is this an empirical observation or provable one? In addition, the proof of Theorem 3.3 does not actually require all Hessians to be degenerate, just that the one used in the hidden fixed point space is nondegenerate. The current assumption that all critical points are degenerate was made to simplify the statement of the theorem but can be changed in the revised version. The assumption on dimensions is the most restrictive one.
>
>      As for practical usefulness, we do also have some results on $S^2$ activations (Appendix C.3), a component used in real equivariant models such as EquiformerV2. There are conditions where spurious minima arise and we sometimes empirically still see such minima even when the dimension condition of Theorem 3.3 is not satisfied.
>
>      Finally, while complaints about applicability of Theorem 3.3 are more justified due to the dimension assumption, we emphasize that Lemma 3.2 is very general and has very mild assumptions. We believe the identification of these hidden symmetry induced critical points is very important even if they are not minima or imply minima (as happens for the additional criteria of Theorem 3.3). This is because saddle points are a known problem for optimization [1].
>
> ## Questions
> * See weaknesses
>
> * This is a great question! Again, we emphasize that the hidden symmetry idea of Section 3.1 is independent of equivariance and could be applied to other types of networks.
>
>      Further, we would like to add that Lemma 3.2 and Theorem 3.3 lays the foundation for future work which may investigate how the number of non-global minimum critical points grows as we overparameterize, similar in spirit to [2]. This could potentially give insight on differences in scaling behavior between constrained and unconstrained models.
>
> * This is a very interesting question! Before answering, we want to emphasize that our work focuses on describing the loss landscape by characterizing critical points and minima rather than navigating optimization on the landscape (eg. gradient descent). The latter has a number of subtleties including a crucial dependence on parameterization [3].
>
>      As for the usefulness of weight decay and gradient normalization, we expect these will not help. If we model weight decay with L2 regularization, we note that the parameter symmetries will still be symmetries of the loss function and so it should not affect our analysis. For gradient normalization, we expect no component perpendicular to the boundary between good and bad regions in the first place and normalizing should not affect this. We also tried using Adam instead of gradient descent and found qualitatively similar results to those in Figure 5 and can add these results in the appendix.
>
> We thank you again for your time in reviewing our work. Please let us know if you have any other concerns.
>
> ## References
>
> [1] Dauphin, Yann N., Razvan Pascanu, Caglar Gulcehre, Kyunghyun Cho, Surya Ganguli, and Yoshua Bengio. "Identifying and attacking the saddle point problem in high-dimensional non-convex optimization." Advances in neural information processing systems 27 (2014).
>
> [2] Simsek, Berfin, François Ged, Arthur Jacot, Francesco Spadaro, Clément Hongler, Wulfram Gerstner, and Johanni Brea. "Geometry of the loss landscape in overparameterized neural networks: Symmetries and invariances." In International Conference on Machine Learning, pp. 9722-9732. PMLR, 2021.
>
> [3] Kristiadi, A., Dangel, F., & Hennig, P. (2023). The geometry of neural nets' parameter spaces under reparametrization. Advances in Neural Information Processing Systems, 36, 17669-17688.

---

> > ### Comment · Reviewer_jLby · 2025-08-03
> >
> > Thanks for your detailed reply. My concerns are addressed and I will therefore maintain my score. I encourage the authors to remove the assumption that all critical points are nondegenerate.
> >
> > > we would appreciate a reference for degenerate Hessians in real GNNs
> >
> > Some citations:
> >
> > - Classics:
> >    - Sagun, Levent, Leon Bottou, and Yann LeCun. "Eigenvalues of the hessian in deep learning: Singularity and beyond." arXiv preprint arXiv:1611.07476 (2016).
> >    - Sagun, Levent, Utku Evci, V. Uğur Güney, Yann Dauphin, and Léon Bottou. "Empirical analysis of the Hessian of over-parametrized neural networks." International Conference on Learning Representations (ICLR), 2018.
> > - More recent:
> >    - New, Alexander, et al. "Curvature-informed multi-task learning for graph networks." arXiv preprint arXiv:2208.01684 (2022).

---

> > > ### Author Response · Authors · 2025-08-04
> > >
> > > We thank you again for your time spent evaluating our work and we truly appreciate your helpful feedback! We also appreciate the references and will update our Theorem 3.3 so the non-degenerate Hessian assumption only applies to the critical point in the hidden fixed point space.

---

### Official Review · Reviewer_Ppai · 2025-07-05

**Clarity:** 4
**Significance:** 3
**Originality:** 4
**Rating:** 5
**Confidence:** 4

**Summary:**

This work investigates the question of whether equivariant constraints in neural networks can create optimization challenges. First, the authors provide an extensive presentation of the preliminary background that is necessary for their analysis. Then, they focus on the analysis of equivariant models that can be derived by constraining the parameters of a fully unconstrained model to a specific parameter subspace. They define the concept of hidden parameter symmetries, which are symmetries in the unconstrained space that disappear in the equivariant subspace, and they show how these can induce critical points or spurious minima in the loss landscape of the constrained equivariant model. Additionally, the authors construct a simple example of a single-layer model where these critical points can be observed experimentally, and they demonstrate how the parameter space can be split into two distinct halves that cannot be traversed by gradient flow. As a result of the theoretical analysis and simple experimental observations, the authors conclude that, indeed, partially removing the equivariant constraints can result in an easier optimization landscape, allowing the network to utilize different types of latent equivariant representations and traverse distinct equivariant parameter subspaces.

**Questions:**

- Can the results of this work be used to provide a guideline on how to relax the equivariant constraint of a given model?
- How reasonable is it to expect that the separation of the parameter space into two halves can also happen in deeper models with a much higher number of parameters?
- How do these critical points induced by hidden symmetries affect optimization if, instead of gradient flow, stochastic gradient descent is used?

**Ethical Concerns:**

["NO or VERY MINOR ethics concerns only"]

**Final Justification:**

I believe that this work provides a novel perspective on possible optimization difficulties in equivariant networks that would be interesting to the community. While the original submission had some issues in the formulation of the theorems, as also pointed out by the area chair, I believe that the authors did a good job resolving them during the rebuttal period. Due to that, and given that the authors will do the appropriate corrections discussed during the discussion with the reviewers and area chair, I keep my rating of accept.

**Limitations:**

Yes

**Paper Formatting Concerns:**

No formatting issues in this paper

**Quality:**

3

**Strengths And Weaknesses:**

Strengths:
- The topic of this work is timely since there are multiple works that showcase evidence supporting the benefits of symmetry breaking in equivariant neural networks. Additionally, while most previous works provide mostly experimental evidence, this paper presents a theoretical hypothesis on the possible mechanism behind this phenomenon.
- The presentation of the theoretical analysis is clear and can be easily understood even by readers who might not be experts in the topic.
- The simple experimental setup provides both evidence supporting the theoretical hypothesis and also helps the readers build a better intuition on the problem at hand.

Weaknesses:
- While the authors provide a possible mechanism showcasing how an equivariant constraint can create optimization difficulties, they do not provide any guidance on how practitioners should relax the equivariant constraints in a way that improves optimization.
- The analysis in this paper utilizes gradient flow for optimization. In practice, most models are trained with much noisier optimization regimes, e.g., stochastic gradient descent. This creates a gap between the theoretical results shown in this paper and what a practitioner expects to see in practice.

---

> ### Author Rebuttal · Authors · 2025-07-31
>
> We thank the reviewer for their positive evaluation of our work! We appreciate that the reviewer finds our work timely, the presentation clear, and the experimental setup supportive and insightful. We hope our response addresses the weaknesses and questions raised.
>
> ## Weaknesses:
> * Thanks for this feedback! Indeed the current work focuses on identifying an interesting mechanism (hidden symmetries) which generates critical points and even spurious minima in constrained models (e.g. equivariance). While providing explicit guidance on relaxation techniques is beyond the scope of our work, we can provide suggestions for mitigating the effects of the spurious minima.
>
>      In practice, we expect increased width to overcome the spurious minima issues in constrained networks. For wide networks, we expect there to be enough transitive blocks initialized on the “good” side so a network could learn to only use the “good” blocks and ignore the “bad” blocks. Further, it turns out if a previous layer is wide, the dimension condition of Theorem 3.3 is less likely to hold.
>
> * We would like to clarify that the theoretical portion focuses on the loss landscape which is independent of optimization methods. Figure 2 depicts a gradient argument to give intuition for Theorem 3.3 but the proof does not rely on gradients. Assuming a nice parameterization, we do not expect SGD to cross the hidden fixed point space. This is because the loss landscape generated from a subset of samples is still subject to the same parameter symmetries which generated the fixed point space. Whether SGD can help for other parameterizations (equivalently changing learning rates on individual parameters) is an interesting question. However, we still expect that a network gets stuck when initialized too close to a bad minima.
>
>      In addition, we ran some experiments using Adam instead of gradient descent and obtained similar results to Figure 5 already in the paper.
>
> ## Questions
> * See weaknesses 1
> * See weaknesses 1. We also want to add that the hidden symmetry mechanism described for permuting neurons applies to every layer so we expect increasing depth to have minimal impact on those results.
> * See weaknesses 2
>
>      Finally, we would like to add that Lemma 3.2 and Theorem 3.3 lays the foundation for future work which may investigate how the number of non-global minimum critical points grows as we overparameterize, similar in spirit to [2]. This could potentially give insight on differences in scaling behavior between equivariant and nonequivariant models.
>
> We thank you again for your time in reviewing our work. Please let us know if you have any other concerns.
> ## References
>
> [1] Dauphin, Yann N., Razvan Pascanu, Caglar Gulcehre, Kyunghyun Cho, Surya Ganguli, and Yoshua Bengio. "Identifying and attacking the saddle point problem in high-dimensional non-convex optimization." Advances in neural information processing systems 27 (2014).
>
> [2] Simsek, Berfin, François Ged, Arthur Jacot, Francesco Spadaro, Clément Hongler, Wulfram Gerstner, and Johanni Brea. "Geometry of the loss landscape in overparameterized neural networks: Symmetries and invariances." In International Conference on Machine Learning, pp. 9722-9732. PMLR, 2021.

---

> > ### Comment · Reviewer_Ppai · 2025-08-07
> > **Response to Rebuttal**
> >
> > I thank the authors for their rebuttal and for addressing my concerns about this work.
> > First, I appreciate the addition of possible suggestions that can mitigate the identified optimization problem.
> > Second regarding weakness 2, while I recognize that this work mainly focus on the loss landscape, it still cannot be completely independent to the optimization method uses. Because by describing a loss landscape as problematic we make the implicit assumption that it is problematic because it doesn't allow a specific optimization method to converge to a "good" solution. Thus I would appreciate a more detailed discussion about how the choice of optimization methods affect or not the identified problem.
> > Since the authors addressed most of my concerns I will keep my rating of accept.

---

> > > ### Author Response · Authors · 2025-08-08
> > >
> > > We thank you again for your time spent evaluating our work and we truly appreciate your helpful feedback!
> > >
> > > We agree that different optimizers perform differently on the landscape and affects the characterization of how "bad" the landscape is. If accepted, we will also add a more in depth discussion of why the issues described has an impact on SGD, as presented in the rebuttal, in addition to the gradient argument in Figure 2. We can also add additional plots similar to Figure 5 where we swap gradient descent with other common optimizers such as Adam to illustrate how the identified issue still persists.

---

### Official Review · Reviewer_cpcL · 2025-07-07

**Clarity:** 2
**Significance:** 3
**Originality:** 3
**Rating:** 5
**Confidence:** 4

**Summary:**

The paper investigates the consequences for considering the loss-landscape of designing neural networks to be equivariant. This is done with an MLP as a case study. The authors adopt the view that equivariant networks should be seen as part of a larger family of un constrained networks. In this case it turns out that the loss landscape of the equivariant networks may have minima which are saddle points in the case of unconstrained networks; a feature which would lead to undesirable training behavior.

In general the idea is interesting and well founded. I thus recommend that the paper is accepted.

**Questions:**

- Line 24-25: I assume it is an error that this sentence is included?
- Line 85: I suggest that the authors provide a reference to how maximally expressive equivariant linear layers are constructed forn an irrep decomposition.
- Line 94: Is it assumed implicitly that G is finite?
- Line 105: Isn't it meant to be $\rho \circ \pi: G \to GL(V_n)$ instead of $\rho: G \to GL(V_n)$ to be consistent with previous notation?
- Line 107. I assume that the action of $G$ on the set $\{b_, ..., b_n\}$ is transitive if it only have one orbit?
- Line 109: Please provide a reference or a proof for this Lemma.
- Line 128: Why is it needed that $p$ is a homeomorphism (wrt. the Euclidean topology i assume)? Is it not sufficient to require it to be bijective for P to be a group?
- Line 152: There seems to be something wrong with the sentence "Then if $\Sigma is ...$".
- Line 179: By linearly constrained subspace do you mean a fixed point subspace?
- Line 180: Which group is $p$ an element of and what is the action?
- As mentioned, a main point of the paper, as I understand it, is that equivariant networks might introduce minima, which would not appear in a corresponding unconstrained model. However do you have any reflections on whether this poses a problem empirically? I ask since often networks will be  extremely overparametrized, and we rarely seek a global minima anyway. As such one could think that the additional minima could be introduced through the use of an equivariant model would not pose a problem.

**Ethical Concerns:**

["NO or VERY MINOR ethics concerns only"]

**Final Justification:**

The paper addresses an interesting and relevant problem, and the authors provide a principled approach to understanding the topic. The weaknesses I observed in the paper was mainly related to communication of some theoretical results. However, the authors were able to mitigate this weakness during the discussion and rebuttal phase.

**Limitations:**

Yes.

**Paper Formatting Concerns:**

No observed concerns.

**Quality:**

3

**Strengths And Weaknesses:**

Strengths:
- The idea is interesting and compelling. I applaud that the authors for writing a paper aimed at principled understanding of a topic.
- The manuscript is in general well written.
- The authors comprehensively reflect on the limitations of their method.

Weaknesses:
- The paper has a variety of central results (Theorem 3.3, 3.4 and 3.5). Unfortunately the proofs of all these important results are moved to the appendix. I would suggest to the authors that they provide either the full proof or a sketch of proof in the main body of the paper.
- I am slightly confused about section 3.2 overall. As I understand it, it is supposed to characterize hidden fixed point spaces of equivariant networks. However, I do not see the direct connection to the previous sections.
- I urge the authors to work on the clarity in communication of the ideas proposed. For instance a main point of the paper, as I understand it, is that equivariant networks might introduce minima, which would not appear in a corresponding unconstrained model. I suggest that the authors to attempt to make it even more clear how this happens and why it is a problem, e.g. through further use of illustrations or examples.

---

> ### Author Rebuttal · Authors · 2025-07-31
>
> We thank the reviewer for their insightful comments. We appreciate that the reviewer finds our work compelling, principled and well written. We hope our response addresses the weaknesses and questions raised.
>
> ## Weaknesses:
> * We agree that it would be nice to include proof sketches but were unable to due to space constraints. Instead, we provided Figure 2 giving an intuitive argument for why Theorem 3.3 is reasonable and Figure 3 which tries to visually explain the results of Theorems 3.4 and 3.5. We felt this provided a reasonable balance especially since some of the proofs are rather mechanical. However, we are open to further suggestions!
> * Thanks for this feedback! The high level idea is that the fixed point subspace for a parameter symmetry of permuting two neurons corresponds to those two neurons having the same input and output weights. From Lemma 3.2 and Theorem 3.3, we care about how this fixed point space intersects with an equivariant subspace. What Theorems 3.4 and 3.5 show is that within an equivariant subspace, equivalent neurons cannot happen in isolation. Instead, the existence of a single pair of equivalent neurons must imply a number of other neurons also be equivalent in a structured way as depicted in Figure 3.
> * Thanks for this suggestion! If accepted, we will add figures to motivate/describe the hidden symmetry idea which underlies our main results. We hope the reviewer finds the existing Figure 2 insightful for motivating Theorem 3.3.
>
>      In addition, we tried to highlight the key takeaways in the current manuscript at the end of both the abstract and introduction. Upon reflection, we believe the following modifications for the first 2 points will make the ideas more clear.
>
>      * By viewing a given architecture as a constrained version of a larger function class, we prove that the previously hidden parameter symmetries of the larger class can imply the existence of critical points and even spurious minima for the constrained network
>      * Hidden symmetries induce multiple symmetrically related but different constrained subspaces. Empirically, we find a spurious minimum in an equivariantly constrained subspace is often a saddle in the unconstrained space and escapes to a better minimum in a symmetrically related subspace.
>
> ## Questions:
> * Line 24-25: This is meant to be a parody of the famous first sentence of the Tale of Two Cities novel which inspired our title. This can be removed if the reviewer feels strongly.
> * Line 85: Good suggestion! We will add references to standard group theory texts for Schur’s lemma. In addition, we will add a discussion of Schur’s lemma in the group theory section of the appendix.
> * Line 94: It is not required that $G$ be finite, however the permutation representations are finite in this definition. In principle, we can extend the definition to cover infinite dimensional representations by replacing $\mathrm{Sym}(n)$ with $\mathrm{Aut}(X)$ for some object $X$ in Definition 2.4. We can add a discussion of this extension in the appendix.
> * Line 105: Yes, although we would like to only use $\rho$ to simplify the writing. We will make sure to clarify this in the revised version.
> * Line 107: Yes, exactly!
> * Line 109: We did put this in the appendix (but forgot to fix the numbering). It starts on Line 545 or also Lemma A.15.
> * Line 128: You are right that bijection is all we need. We originally wanted to only consider “nice” transformations which is why we chose homeomorphisms.
>
>      However, upon closer inspection, we realize that we assumed linear transformations in our main theorems without mentioning it and will update our manuscript to include this. This is not a major restriction as most commonly considered parameter symmetries (permutations, scaling) are linear transformations.
> * Line 152: We believe changing this to “If $\Sigma$ is …” should fix this
> * Line 179: No, we mean a subspace satisfying some other linear constraints (e.g. equivariance constraints)
> * Line 180: Thanks for pointing this out! We meant for $p$ to be a parameter symmetry of the unconstrained networks.
> * This is a good question! We do indeed expect the spurious minima problem to mostly disappear as we increase width. For wide networks, we expect there to be enough transitive blocks initialized on the “good” side so a network could learn to only use the “good” blocks and ignore the “bad” blocks. Further, if a previous layer is wide, the dimension condition of Theorem 3.3 is less likely to hold.
>
>      Finally, while the dimension condition of Theorem 3.3 restricts its applicability for wider networks, we emphasize that Lemma 3.2 has very mild conditions and will still imply hidden symmetry induced critical points! This is still very important because saddle points are a known problem for optimization [1]. Further, this Lemma lays the foundation for future work which may investigate how the number of non-global minimum critical points grows as we overparameterize, similar in spirit to [2]. This could potentially give insight on differences in scaling behavior between equivariant and nonequivariant models.
>
> We thank you again for your time in reviewing our work. Please let us know if you have any other concerns.
>
> #References
>
> [1] Dauphin, Yann N., Razvan Pascanu, Caglar Gulcehre, Kyunghyun Cho, Surya Ganguli, and Yoshua Bengio. "Identifying and attacking the saddle point problem in high-dimensional non-convex optimization." Advances in neural information processing systems 27 (2014).
>
> [2] Simsek, Berfin, François Ged, Arthur Jacot, Francesco Spadaro, Clément Hongler, Wulfram Gerstner, and Johanni Brea. "Geometry of the loss landscape in overparameterized neural networks: Symmetries and invariances." In International Conference on Machine Learning, pp. 9722-9732. PMLR, 2021.

---

> > ### Comment · Reviewer_cpcL · 2025-08-07
> > **Answer to Rebuttal**
> >
> > Thank you for providing a thorough rebuttal, and addressing the raised concerns. Having considered your rebuttal as well as the answers provided to the rest of the reviewers I have decided to increase the score of the paper.

---

> > > ### Author Response · Authors · 2025-08-08
> > >
> > > We thank you again for your time spent evaluating our work and we truly appreciate your helpful feedback!

---

### Comment · Area_Chair_mkg8 · 2025-08-01

Dear authors,

Thank you for submitting your work and for the effort invested in preparing it. The reviews raise several points for discussion regarding your paper. Some of the concerns raised, particularly those noted by Reviewer AZWo, suggest that providing a clearer articulation of your contributions could help reviewers reach a more accurate assessment. A concise bullet-point summary highlighting the main technical contributions and the novelty of your results may help clarify their distinctive aspects within the broader landscape.

In addition, since global and spurious minima in the exact teacher-student setting analyzed in your work have already been examined in related literature, a brief comparison to:

[a] *Dynamics of stochastic gradient descent for two-layer neural networks in the teacher-student setup* (Goldt et al.),

[b] *Analytic Study of Families of Spurious Minima in Two-Layer ReLU Neural Networks: A Tale of Symmetry II* (Arjevani & Field),

may help better understand how your findings complement existing results.

Finally, as repeatedly noted in the reviews, clarifying how the empirical section illustrates various practical scenarios where the co-dimension one case (or the assumptions of Lemma 3.2) arise, and how these relate to your observations and results, could help reviewers more accurately assess the scope and implications of your work.

Thank you for your thoughtful contribution to the discussion,\
AC

---

> ### Author Response · Authors · 2025-08-02
> **Comparison with other student-teacher studies**
>
> We thank the area chair for these suggestions and bringing up other related work. We separately posted a general comment highlighting our main contributions and discussing the practicality/limitations. We primarily focus on comparison to other works on teacher-student setup in this response.
>
> As pointed out, the two-layer teacher student setup has been the subject of numerous studies. However, the focus of our work is not on understanding such toy models but rather **using them to illustrate the main theoretical mechanism** described in our work. As such, we believe our work is quite distinct from such studies.
>
> However, works focused on two-layer teacher student setups do contain many interesting ideas which complement some of our own.
>
> In [1], the focus is on understanding SGD dynamics in two-layer networks. In particular, they derive a very interesting set of differential equations to describe training behavior. However, this is quite different from our goal of understanding loss landscapes. Further, our training setup is a bit different as we employ weight sharing to enforce equivariance constraints.
>
> In [2] (and a series of related works by the same authors), the key goal is understanding minima of two-layer ReLU networks. In particular, they introduce a very interesting symmetry based technique to characterize families of minima. However, our understanding is that the teacher network in their setup is fixed, with the identity as the first weight matrix and a constant 1 matrix as the second weight. This affords a nice comparison since their teacher network is equivariant and our Theorem 3.3 also applies.
>
> In particular, we might expect an additional minima in the constrained space where the diagonal value is less than the off diagonal, this seems to correspond to the family 1(b) in Theorem 1 of [2]. Similarly, by considering equivariance to subgroups of $S_d$ and invariance of the input distribution to $S_d$, our Theorem 3.3 would also imply existence of the other families. Of course, what is powerful about their Theorem 1 is it not only shows existence but completely characterizes what these minima look like.
>
> While a very impressive and comprehensive characterization, it is unclear to us how easily the techniques of [2] generalize to other architectures and less symmetric teacher weights. In contrast, our Lemma 3.2 and Theorem 3.3 package the hidden symmetry idea into a relatively general statement.
>
> ## References
>
> [1] Goldt, Sebastian, Madhu Advani, Andrew M. Saxe, Florent Krzakala, and Lenka Zdeborová. "Dynamics of stochastic gradient descent for two-layer neural networks in the teacher-student setup." Advances in neural information processing systems 32 (2019).
>
> [2] Arjevani, Yossi, and Michael Field. "Analytic study of families of spurious minima in two-layer relu neural networks: a tale of symmetry ii." Advances in Neural Information Processing Systems 34 (2021): 15162-15174.

---

> > ### Comment · Area_Chair_mkg8 · 2025-08-06
> >
> > We thank the authors for the detailed and helpful response. Several issues remain. The message focuses on some of the main technical contributions stated by the authors: Lemma 3.2, Theorem 3.3, and the relationship of the experimental section to the latter.
> >
> > ---
> >
> > ## Lemma 3.2
> >
> > 1. Could you please elaborate on your statement:
> >
> > "In addition, we forgot to include the geometric condition required in Lemma 3.2. We will add this in the revised version."
> >
> > Specifically, are the proof and statements of Lemma 3.2 (and the related results) correct as given?
> >
> >
> > 2. **Issues in the proof**. The final transition in the chain of equations starting at line 568 is incorrect. The group element $h$ is used twice—once as a fixed element and once as the summation index. Even if replaced by a different summation index, the transition remains unclear: why are the two expressions (application of $h$ and projection) equivalent?
> >
> > 3. **Issues in the proof**. Schur orthogonality relations concern characters (not mentioned in the paper). Did you perhaps mean the Reynolds operator?
> >
> >
> > 4. **Practical Relevance.** Lemma 3.2 is described as a main technical contribution but, aside from the concluding remarks, is only cited in the proof of Theorem 3.3, with no further discussion of its scope or use. Moreover, since it is given as an immediate consequence of the basic fact of equivariance of gradients of invariant functions, its technical novelty is unclear. Therefore, could you please clarify its standalone significance by providing examples or applications where its assumptions hold, and describe the significance of the outcome in such cases?
> >
> >
> > ---
> >
> > ## Theorem 3.3:
> >
> > 5. **Concerns regarding the proof**. Is the codimension-one condition the only assumption used in the proof to apply Lemma 3.2? If that’s the case, could you explain why the example below isn’t a counterexample to the argument?
> >
> > Take $\Theta = \mathbb{R}^2$, $\tilde{\Theta} = \\{ (x, 0) | x\in \mathbb{R} \\}$, $H=S_2$ with the natural action.
> >
> > Thus,
> > $\Theta_{H} = \\{(x,x)|x\in\mathbb{R}\\}$,
> > so
> > $\tilde{\Theta}_{H} = 0$ is co-dimension one in $\tilde{\Theta}$.
> >
> > However, $\Theta_{H}^\perp = \\{(x,-x)|x\in\mathbb{R}\\}$ and so
> > $\Theta_{H}^\perp \cap \tilde{\Theta}$ is also the zero vector space. Thus,
> > $(\Theta_{H} \cap \tilde{\Theta}) \oplus (\Theta_{H}^\perp \cap \tilde{\Theta}) = 0 \neq \tilde{\Theta}$, in violation of the hypotheses of Lemma 3.2.
> >
> > Multiply the construction by different factors (represnations) to obtain more “interesting” examples and higher dimensionality.
> >
> > 7. **Scope and MLPs**. Referring to the authors’ statement, “Most assumptions of Theorem 3.3 are mild,” could you confirm the following points regarding the scope of the theorem as stated in the paper?
> >
> > a. **No degenerate Hessians and MLPs**. If even training only the second layer is allowed, all Hessians become degenerate (see [b]), an immediate consequence of the respective Lie group action. Does this imply that Theorem 3.3 is inapplicable to essentially all MLPs with more than one layer?
> >
> > b. **No degenerate Hessians and MLPs**. The authors mention that “when degeneracy occurs, it likely results from some symmetry which can be exploited to preserve the theorem.” Could the authors please provide a precise statement and a proof supporting this claim? Without this, the applicability of Theorem 3.3 to the setting it aims to address, MLPs, becomes invalid.
> >
> > c. Even if the second layer is fixed but a different activation function is employed, Hessians remain degenerate at certain points.
> >
> >
> > d. The statement "This is reasonable as the set of degenerate symmetric matrices has measure 0" is not generally correct for G-invariant functions without further assumptions. While degenerate symmetric matrices are of measure zero within the space of all symmetric matrices, this property does not necessarily hold for Hessians associated with G-invariant functions. Please clarify or revise this point accordingly.
> >
> > e. Co-dimension assumption: please note that all the fixed point spaces of the minima characterized in [c] (including those of $\Delta S_d$) have co-dimension greater than 1.
> >
> > 7. **Scope**. While the authors “emphasize that Section 3.1 is very general,” they also acknowledge that the “co-dimension assumption is quite strong.” This raises questions about the overall generality of section 3.1: the applicability of Theorem 3.3 is limited by this assumption (and see below), and, as noted, Lemma 3.2 is used only within the proof of Theorem 3.3. Could you please clarify the relationship between these statements?
> >
> > 8. **Practical relevance**. As with Lemma 3.2, and considering that the assumptions of Theorem 3.3 have not yet been verified in the experimental section, could you please provide examples from applications where these assumptions are known to hold?

---

> > > ### Comment · Area_Chair_mkg8 · 2025-08-06
> > >
> > > ## Experimental Section
> > >
> > > 9. Since the teacher-student setting you use is intended to demonstrate Theorem 3.3, could you please provide a proof that the following hypotheses hold:
> > >
> > > - $C$ such that $\tilde{r}\cdot \nabla L(C\tilde{r}) > 0$
> > > - Hessians are nondegenerate
> > > - In particular, is the loss function $C^1$?
> > >
> > > 10. We thank the authors for bringing attention to the body of work related to [b], cited earlier, which we have examined carefully. The fixed point spaces for the permutation representation in the teacher-student setting described in the paper have been studied in (Arjevani & Field 2019), particularly examining different target matrices and distributions, and isotropy groups corresponding to intransitive and transitive primitive subgroups.
> > > In addition, Figure 6 depicts local minima corresponding to families 1 and 2 in [c]. Please clarify the relationship between the setting used in your work and that of the prior work in the revised version. If you consider there to be substantive differences, kindly make these distinctions explicit in the text.
> > >
> > > 11. Continuing the previous point, the statement “For our experiments, we consider a teacher-student setup so we can guarantee perfect loss is achievable” appears. Why is this assumption necessary? Is it a requirement of theorem 3.3?
> > >
> > > ## Presenation (partial)
> > >
> > > - Lemma 2.6: Where is it used or referred to in the paper?
> > >
> > > - $\theta_1$ and $\theta_2$ are used both as coordinates and to denote critical points in $\Theta$.
> > >
> > > ---
> > > ## References
> > >
> > > [b] Tale of Symmetry II (NeurIPS 2021)\
> > > [c] Tale of Symmetry I (NeurIPS 2020)

---

> > > > ### Author Response · Authors · 2025-08-07
> > > > **Response part 1 (Lemma 3.2)**
> > > >
> > > > We sincerely thank the area chair for their thoughtful reading of our paper and for raising additional insightful questions.These have been very helpful for enhancing the presentation of our results and for thinking about their relation to other works.
> > > >
> > > > ## Lemma 3.2
> > > > 1. > In addition, we forgot to include the geometric condition required in Lemma 3.2. We will add this in the revised version.
> > > >
> > > >      We believe there is a misunderstanding. Lemma 3.2 is correct as written. What we meant is that Theorem 3.3 also needs the geometric assumption $(\Theta_H\cap\tilde{\Theta})\oplus(\Theta_H^\perp\cap\tilde{\Theta})=\tilde{\Theta}$ necessary for Lemma 3.2 to apply.
> > > >
> > > >      One additional note, we realize that this condition is equivalent to the cleaner condition $\Pi_H\tilde{\Theta}\subseteq\tilde{\Theta}$ where $\Pi_H$ is the orthogonal projection onto the fixed point space of $H$. We are considering replacing the statement of the condition with this.
> > > >
> > > > 2. In line 568, we first showed $\nabla L(\theta)=h\nabla L(\theta)$ for all $h\in H$. This means $\sum_{h’\in H}h’\nabla L(\theta)=\sum_{h’\in H} \nabla L(\theta)=|H|\nabla L(\theta)$. Dividing by $|H|$ then gives the last equality in line 568. We omitted writing this intermediate step as we felt it was straightforward but can add it if requested.
> > > >
> > > > 3. We specifically were referring to the coordinates version which states for any irreps $\Gamma^{\lambda},\Gamma^{\mu}$ that $\sum_{h\in H}\Gamma_{ij}^{\lambda}(h)\Gamma_{i’j’}^{\mu}(h)=\delta_{\lambda\mu}\delta_{ii’}\delta_{jj’}\frac{|H|}{d_\lambda}$ where $d_{\lambda}$ is the dimension of the irrep. Replacing $\lambda^\mu$ with the trivial representation, this immediately implies $\sum_{h\in H}\Gamma^{\lambda}_{ij}(h)=0$ for nontrivial irreps and gives $|H|$ for the trivial one. Therefore when dividing by $|H|$ the summation gives a projection onto the trivial representation. This particular line of reasoning is well known and can even be found in the Wikipedia page for Schur orthogonality relations.
> > > >
> > > >      In hindsight, the use of the orthogonality relations may be overkill and showing that group averaging is a projection operator can be done with more elementary arguments.
> > > >
> > > > 4. This is a fair question. Our exploration was actually first driven by first observing some of the interesting experimental results. Our efforts to explain the phenomenon led to a focus on Theorem 3.3 with Lemma 3.2 as an intermediate step. It was only in hindsight that we now realize Lemma 3.2 is also quite interesting.
> > > >
> > > >      For applicability, the conditions are satisfied for any linear layer built with permutation representations. This follows from a slight modification of our proof of Theorem 3.4. In particular, Theorem 3.4 is proved by applying weight averaging (corresponds to averaging over a group A) and careful book-keeping. But the weight averaging corresponds exactly to projecting to an appropriately chosen hidden fixed point space. In addition, please see our last point in the general response for our view on how it may help understand overparameterization effects.
> > > >
> > > >      Finally, the **technical novelty comes from the hidden symmetry idea**. If we restricted ourselves to the domain of $\tilde{\Theta}$, **we would never consider the hidden symmetry** $H$ since such transformations are not automorphisms of $\tilde{\Theta}$ but only of the unconstrained $\Theta$. It is only after realizing we must expand our domain to $\Theta$ that we can use standard equivariance of gradient arguments.

---

> > > > > ### Author Response · Authors · 2025-08-07
> > > > > **Response part 2 (Theorem 3.3 part 1)**
> > > > >
> > > > > ## Theorem 3.3
> > > > > 5. See point 1. We believe there is a misunderstanding. Your counterexample is correct and we mean that Theorem 3.3 is missing the additional condition required for Lemma 3.2.
> > > > > 6.
> > > > >      **a.** Can you elaborate on the Lie group action considered. Is this referring to scaling symmetry present in ReLU? Or is this related to additional symmetry in output weights (only the sum matters) when input weights to a pair of neurons are the same? In either of these cases, a L2-regularization term on the weights removes the degeneracy. Further, the latter does not happen if we expect to extract a trivial representation since all output weights would be tied together anyway.
> > > > >
> > > > >      Further, **we wonder if there is a confusion between degenerate critical points and degenerate eigenvalues**. The former is the condition we need and requires the Hessian to be non-singular. The latter is what is exploited using symmetry arguments in the works [2,3].
> > > > >
> > > > >      **b.** We will provide an example and argument sketch. We believe a full proof beyond the scope of this work and we explicitly mention it as a possible future extension (line 315-316).
> > > > >
> > > > >      When making this statement, we were specifically thinking of the case where input weights to a pair (or more) of neurons are the same. Here, the sum of output weights of those neurons is all that matters so there is additional symmetry. In this case, both the co-dimension and non-degenerate Hessian assumptions break.
> > > > >
> > > > >      First, for the co-dimension assumption, it breaks because the hidden fixed point space would require output weights to be shared as well. However, because the sum of output weights of identical neurons is all that matters (see irreducible neurons idea [1]), we can relax the sharing constraint on output weights without changing the function class (by utilizing a local parameter symmetry). Hence we have expanded the space used to divide the constrained subspace. Call it the expanded fixed point space. The new condition would be for this expanded fixed point space to have co-dimension 1.
> > > > >
> > > > >      Next, for Hessian, it is degenerate because there must be 0-eigenvalues corresponding to eigenvector directions not changing the sum of output weights. However, these directions do not matter as they make us stay in the expanded fixed point space. What we care about are directions not in the expanded fixed point space. If in these other directions we have nonzero eigenvalues, then we can either conclude the critical point is a minima or has at most 1 negative eigenvalue. The rest of the argument for Theorem 3.3 would follow in the same way.
> > > > >
> > > > >      **c.** As we pointed out in the general response, we really only require non-degeneracy for the critical point considered in the hidden fixed point space and not all. As suggested by reviewer jLby, we plan to revise the theorem statement to reflect this.
> > > > >
> > > > >      **d.** Fair question. All we wanted is to emphasize the unlikelihood of having eigenvalues of 0 without some underlying reason. See point b for an outline of how one might work around degenerate Hessians when the degeneracy is due to some symmetry. Also, Figure 6 of [3] seems to support our assumption.
> > > > >      **e.** We believe this is a misunderstanding of how we would use Theorem 3.3 in this context. We expand on it here for their family 1(a) minima.
> > > > >
> > > > >      First, we consider the constrained subspace where the first layer weights is spanned by $I,\mathbf{11}^T-I$ and last layer is spanned by $\mathbf{1}^T$. This corresponds to an architecture equivariant to $S_d$ permutation action on the inputs. Note the target weights of $I$ for first layer and $\mathbf{1}^T$ in the last layer is in this constrained subspace. Next, consider the parameter symmetry corresponding to permuting all neurons in the hidden layer. This gives a fixed point space spanned by $\mathbf{11}^T$ in the first layer and by $\mathbf{1}^T$ in the last layer, which has co-dimension 1 with respect to the original constrained space. Our Theorem 3.3 implies there must be an additional minimum in the constrained space either in the fixed point space or with diagonal value less than the off diagonal value. The latter case is what actually happens. Finally, we note that the unit normal data distribution is invariant to the $S_d$ permutation action. Hence, the results of [4] imply that this additional minimum in the constrained space must also be a critical point of the unconstrained space. While the combination of these results cannot imply that these additional critical points are in fact minima in the unconstrained space (as proven in [2] for large enough dimensions), it does make it clear that such points make sense as candidates.
> > > > >
> > > > >      Finally, all of the above analysis was not the original goal of our work. Our focus is on understanding constrained networks. We just find it interesting that our work combined with [4] gives results even on unconstrained networks that fits nicely with the known results of [2,3].

---

> > > > > > ### Author Response · Authors · 2025-08-07
> > > > > > **Response part 3 (Theorem 3.3 part 2)**
> > > > > >
> > > > > > 7. Section 3.1 introduces the idea of hidden symmetries which applies for any model which can be viewed as a constrained version of a larger class. This is a very general idea as it makes no assumptions on model type. Lemma 3.2 tells us the hidden parameter symmetries of the unconstrained class can induce critical points, analogous to previous works using normal parameter symmetries to generate critical points. Lastly, Theorem 3.3 goes a step further and shows in some restricted scenarios, the hidden symmetry induced critical points also induce minima. Our lack of emphasis on Lemma 3.2 was due to our efforts to understand the experimental results which Theorem 3.3 describes quite nicely. Only later did we realize that Lemma 3.2 is also interesting.
> > > > > >
> > > > > > 8. Please see 9. The non-degenerate Hessian assumption is what we cannot prove but see our points above on what part of that assumption is really necessary. Further, the existing Hessian characterization of [2,3] seems to support our assumption of no 0-eigenvalues.

---

> > > > > > > ### Author Response · Authors · 2025-08-07
> > > > > > > **Response part 4 (Experimental section + Presentation)**
> > > > > > >
> > > > > > > ## Experimental Section
> > > > > > > 9. The point of our experimental section is to demonstrate the consequences implied by Theorem 3.3 can realistically appear, not to verify that the assumptions required happen. If we can prove the latter, then there is no need for experiments to show an already proven result.
> > > > > > >
> > > > > > >      In fact, we believe experimental results are more interesting if they agree with theory even without satisfying the necessary assumptions.
> > > > > > >      * In our setting, this is actually not satisfied because there can be large weights which cancel hence not giving strictly positive gradients. This can be fixed by simply adding a L2-regularization term. Please see our discussion with Reviewer Xb7d.
> > > > > > >      * At the moment, we cannot prove whether this condition is satisfied for the needed critical point. The eigenvalue spectra found in [3] shows all nonzero values which seems to support our assumption.
> > > > > > >      * If using the known analytic loss for a unit normal input distribution, it is smooth for almost all points [5].
> > > > > > >
> > > > > > >      Despite this, we observe multiple minima as expected from Theorem 3.3. Perhaps we should reword our statements in the experimental section. We verify the co-dimension assumption and because the other conditions seem milder but harder to verify, we should state we expect Theorem 3.3 to apply, rather than claiming Theorem 3.3 applies.
> > > > > > >
> > > > > > > 10. First, the fixed point subspaces studied in [2] are more similar to equivariance constraints than the fixed point subspaces from parameter symmetries we consider. Their target function is set to have identity first layer weights and all 1’s in second layer, which is equivariant to $S_d$. The other spaces considered in [2] essentially correspond to only enforcing equivariance to subgroups such as $S_{d-1}$ or $S_{d-2}\times S_2$.
> > > > > > >
> > > > > > >      > particularly examining different target matrices and distributions
> > > > > > >
> > > > > > >      It is unclear to us where this statement comes from. Their paper makes it clear that the target matrices are the identity [2] (or orthogonal in [3] which has identical Hessian spectrum as the identity case). Further, they make it clear that the input distribution is unit normal gaussian. The unit gaussian distribution is standard for this type of 2-layer teacher-student setup.
> > > > > > >
> > > > > > >      Our experimental setup is mostly standard and commonly studied [5, 6]. What distinguishes our setup is the enforcement of equivariance constraints. While [2,3] can be viewed as equivariant, the allowed architecture and target weights is quite restricted (only identity) compared to our setup. This restriction is made in [2,3] to make characterizing Hessian spectra a tractable problem. Further, it seems [2,3] never plots loss landscapes.
> > > > > > >
> > > > > > > 11. This is not a necessary assumption. We wanted to come up with a toy experiment that seems like an ideal scenario for equivariant networks: where the equivariant network can exactly learn the target function and the distribution is invariant to the given data symmetry.
> > > > > > >
> > > > > > > ## Presentation
> > > > > > > * Good question. We present this as background material. We believe it is an interesting result that helps in understanding how transitive permutation representations are categorized.
> > > > > > > * Fair point, we will clarify the notation.
> > > > > > >
> > > > > > > We hope this response helps clarify your questions. Please let us know if you have any additional questions or concerns.
> > > > > > >
> > > > > > > ## References
> > > > > > >
> > > > > > > [1] Simsek, Berfin, François Ged, Arthur Jacot, Francesco Spadaro, Clément Hongler, Wulfram Gerstner, and Johanni Brea. "Geometry of the loss landscape in overparameterized neural networks: Symmetries and invariances."
> > > > > > >
> > > > > > > [2] Arjevani, Yossi, and Michael Field. "Analytic study of families of spurious minima in two-layer relu neural networks: a tale of symmetry ii." Advances in Neural Information Processing Systems 34 (2021): 15162-15174.
> > > > > > >
> > > > > > > [3] Arjevani, Yossi, and Michael Field. "Analytic characterization of the hessian in shallow relu models: A tale of symmetry." Advances in Neural Information Processing Systems 33 (2020): 5441-5452.
> > > > > > >
> > > > > > > [4] Nordenfors, Oskar, Fredrik Ohlsson, and Axel Flinth. "Optimization Dynamics of Equivariant and Augmented Neural Networks." Transactions on Machine Learning Research.
> > > > > > >
> > > > > > > [5] Tian, Yuandong. "An analytical formula of population gradient for two-layered relu network and its applications in convergence and critical point analysis." In International conference on machine learning, pp. 3404-3413. PMLR, 2017.
> > > > > > >
> > > > > > > [6] Safran, Itay, and Ohad Shamir. "Spurious local minima are common in two-layer relu neural networks." In International conference on machine learning, pp. 4433-4441. PMLR, 2018.

---

> ### Comment · Area_Chair_mkg8 · 2025-08-08
>
> Since the window for discussion closes soon, I would like to share my impressions briefly and openly. Please read the following in the spirit of constructive feedback.
>
> **You have acknowledged errors in the proof of Lemma 3.2, and you indicate that you would like to restate and change the assumptions.**
>
> - Indeed, the proof is incorrect by items 2 and 3.
> - In fact, the correct reasoning reduces to a straightforward one-line exercise:
>   $x \in V^H \Rightarrow  \nabla f(x)= \nabla f(hx) = h \nabla f(x) \Rightarrow \nabla f(x) \in V^H$.
>
> - Notably, no group averaging is involved here (and you never defined the action on the tangent space).
>
> **You have acknowledged serious errors in Theorem 3.3, both in the statement and in the proof.**
>
> - Here, the proof is not fixable, because the statement is false.
>
> - If you plan in the future to add the condition from Lemma 3.2, then Lemma 3.2 holds, *of course*, but the entire discussion of the codimension-one condition (needed to invoke Lemma 3.2), both in the paper and in the reviews, becomes irrelevant, as does the experiment section in its present form.
>
> **You have acknowledged that although the paper is centered on MLPs, Theorem 3.3 does not apply to networks with more than one trainable layer, and that a fix is beyond the scope of this work.**
>
> - Indeed, already if the second layer is trainable, the degeneracy condition fails.
>
> **You have acknowledged that the assumptions do not hold for the teacher–student setting in the experiment section.**
>
> - In fact, it is not even differentiable everywhere, let alone $C^1$. In particular, the condition relying on $\nabla L(C\theta) > 0$ is trivially false (this is explicitly used in the proof of 3.3), and the degeneracy condition has never been checked.
>
> **You write: “we believe experimental results are more interesting if they agree with theory even without satisfying the necessary assumptions.”**
>
> - Then, what is it that one can reasonably hope to learn from an experiment?
>
> - The two minima, family 1(a) and family 1(b), were detected following *On the Principle of Least Symmetry Breaking in Shallow ReLU Models*, which also studies different targets and distributions. The families of minima shown in the experimental section are **not a contribution of the present work, as the authors have acknowledged.**
>
> ---
>
> It is my considered opinion that the core idea has potential for further development. However, serious flaws are present in every aspect of the current execution. While the five reviewers assigned positive scores, I believe the extensive list of critical issues in execution all but obscures the value of the core idea.
>
> ---
>
> **Given the scale of errors and the revisions required, I wonder if you might consider resubmitting to the next conference after addressing all comments I outlined earlier.**
>
> ---
>
> Note that the list of major issues I have raised is far from exhaustive; given the short time window, I have omitted other serious problems. One additional example of the ones omitted: in your terminology, a “hidden space” is the fixed-point space of a subgroup $H$. In the teacher–student setting, $\tilde{\Theta}$ is clear, but what exactly is $H$? *Your paper does not allow this question to be answered, because the group action is not properly defined.*
>
> In addition, here are a few examples of the writing issues I had to omit:
> - In the literature, “hidden symmetry” refers to a group action that is indeed acting on the space but has yet to be detected.
> - Direct sums are denoted by $\oplus$, not $+$.
> - Fixed-point spaces are denoted $V^H$, not $V_H$.
> - The principle of symmetric critically not used nor needed in this context
> - …

---

> > ### Author Response · Authors · 2025-08-08
> >
> > We believe these remarks are unreasonable and unfounded. Many of these are misinterpretations of our responses above and of our paper. We kindly ask the area chair to carefully reread our responses to your concerns and our discussions with the reviewers.
> >
> > > You have acknowledged errors in the proof of Lemma 3.2, and you indicate that you would like to restate and change the assumptions
> >
> > This is **incorrect**. We are proposing to change to an equivalent assumption which is easier to understand. We emphasize that this is **not a necessary change** and the current assumption is correct as is. Your **claim that the proof is incorrect is unfounded**. We may have overcomplicated one line, this does not mean it is incorrect.
> >
> > > You have acknowledged serious errors in Theorem 3.3
> >
> > We simply forgot to include one additional assumption. **The severity** of the mistake is **greatly exaggerated**. We have discussed in our response above how this assumption holds for the case of permutation representations considered in our paper. **You incorrectly claim** the co-dimension condition is needed for Lemma 3.2. It is a completely independent condition. The discussion of its satisfiability remains relevant.
> >
> > > You have acknowledged that although the paper is centered on MLPs, Theorem 3.3 does not apply to networks with more than one trainable layer
> >
> > **Nowhere do we acknowledge this**. In fact, we specifically point out that if the next layer contains scalar irreps, the degeneracy and codimension conditions hold due to the weight tying that already exists in the architecture. Please understand that we are considering constrained networks and using **equivariant** MLPs as examples, not regular MLPs.
> >
> > > What is it that one can reasonably hope to learn from an experiment?
> >
> > We believe the point of an experiment is to discover interesting phenomena which appear in real life. The point of theory is to understand why such phenomena occur. **To prove anything in theory often requires stronger conditions than can be shown in real life**.
> >
> > > two minima, family 1(a) and family 1(b), were detected following On the Principle of Least Symmetry Breaking in Shallow ReLU Models, which also studies different targets and distributions
> >
> > Please provide the section where that study [1] looks at different targets and distributions. They only consider a target matrix of $I$. Further, the families in those studies [1] are only proven to be minima (in the unconstrained space) for high enough dimensions and could be saddle points at smaller dimensions. We show that in an equivariant subspace, even for smaller dimensions, these are minima. Finally, we consider target matrices other than the identity. This is made quite clear with our explicit depiction of the weights in Figure 6.
> >
> > Lastly, it appears **the Area Chair misses the following important points of our experiments**. We explicitly demonstrate the dividing into two halves behavior in the constrained space (See Figure 5). We also observe interesting change of basis behavior when relaxing equivariance constraints (Figure 6 and 7).
> >
> > > I believe the extensive list of critical issues in execution all but obscures the value of the core idea
> >
> > Based on the positive evaluations from all the reviewers, it appears the core idea is greatly appreciated. **Many explicitly praised the clarity of the theory portion**.
> >
> > > In the teacher–student setting, $\tilde{\Theta}$ is clear, but what exactly is
> > $H$? Your paper does not allow this question to be answered, because the group action is not properly defined.
> >
> > Please see our discussion with Reviewer AZWo. We make it clear what the group action corresponding to permuting two neurons is.
> >
> > > "hidden symmetry" refers to a group action that is indeed acting on the space but has yet to be detected
> >
> > The parameter symmetries considered were indeed not detectable in our context until we decided to expand the parameter space. We believe calling them hidden symmetries is a succinct and accurate description.
> >
> > > Direct sums are denoted by $\oplus$ not $+$
> >
> > It is correct to use $+$ when dealing with sums of subvector spaces. We are not performing direct sums anywhere except for Appendix C.3.
> >
> > > Fixed point spaces are denoted $V^H$ not $V_H$
> >
> > We make this notational choice to avoid writing something like $\Theta^{H,\perp}$.
> >
> > > The principle of symmetric critically not used nor needed in this context
> >
> > This principle is analogous to our Lemma 3.2. We present it as background to motivate why parameter symmetries are helpful.
> >
> > ## References
> >
> > [1] Arjevani, Yossi, and Michael Field. "On the principle of least symmetry breaking in shallow relu models."
> >
> > [2] Arjevani, Yossi, and Michael Field. "Analytic study of families of spurious minima in two-layer relu neural networks: a tale of symmetry ii."
> >
> > [3] Arjevani, Yossi, and Michael Field. "Analytic characterization of the hessian in shallow relu models: A tale of symmetry."

---

### Author Response · Authors · 2025-08-02
**Key highlights and discussion of practicality**

We would like to thank all the reviewers for their time in evaluating our work and the area chair for overseeing the process.

As suggested by the AC we highlight our main technical contributions and novelty. Next, we discuss the practicality of our main results as asked by many reviewers.

## Main technical contributions
* Our key insight is the hidden symmetry idea which drives the results of Lemma 3.2 and Theorem 3.3. As far as we are aware, this idea is novel and potentially a highly generalizable and useful tool.
* Theorems 3.4 and 3.5 appear to be novel. We view these results as related to but separate from works on the expressiveness of permutation representations [2].
* Lastly, we present the novel empirical observation that unconstrained networks can effectively perform a “basis change”; it seems most prior works focus on relaxation methods which return to the same basis.

## Practicality
* The assumption of Lemma 3.2 is a relatively mild but necessary geometric condition corresponding to orthogonality of the fixed point subspace and constrained subspace. This condition will always hold in the case of permutation representations.


* Most assumptions of Theorem 3.3 are mild, but we agree with the reviewers that the co-dimension assumption is quite strong.
     * **Co-dimension assumption:** By far the strongest assumption. If the previous layer has large width, it is unlikely to hold. Where we could see consequences is in the first layer. This is because the input layer width is constrained by our data. We find it unsurprising this condition is less likely for larger networks as there are many more directions where the Hessian can be negative.

     * **Minima not in $\tilde{\Theta}_H$:** Mild assumption. Because the hidden fixed point space has lower dimension and hence measure 0, we believe it is unlikely that all minima are in $\tilde{\Theta}_H$.

     * **$C$ such that $\hat{r}\cdot\nabla L(C\hat{r})>0$:** Mild assumption. This just says if we go in any direction of parameter space far enough, the loss will keep increasing. This is reasonable as very large parameter values will give divergent regularization terms. Further, the purpose of this assumption is to show we can draw a region with no boundary critical points, which can happen even if the assumption is false.

     * **No degenerate Hessians:** Mild assumption. All we need is the Hessian of the critical point in the hidden fixed point space to be non-degenerate. This is reasonable as the set of degenerate symmetric matrices has measure 0. When we have degeneracy, it likely results from some symmetry which we can exploit to preserve the theorem. We assume all critical points have non-degenerate Hessians to simplify the Theorem statement.

     * In addition, we forgot to include the geometric condition required in Lemma 3.2. We will add this in the revised version.

* We emphasize that **Section 3.1 is very general**. All it needs is a linearly constrained class of models and parameter symmetries with linear actions. Since this work was originally motivated by understanding relaxation of equivariant models, the remainder of the work focuses on equivariant networks built with permutation representations.

* Permutation representations are used as **components of real models** such as Deep Sets and group convolutions. Further, for rotation equivariant models, $S^2$-activations are a useful component (notably in EquiformerV2). $S^2$ signals can be viewed as a generalization of a permutation representation. We already have experimental results in Appendix C.3 for $S^2$ signals.

* Many reviewers also ask about the effects of overparameterization. As noted above, we indeed expect overparameterization, particularly on width, to largely overcome the spurious minima issue of Theorem 3.3. However, **Lemma 3.2 still applies** and the implied critical points are analogous to the symmetry induced critical points of [3]. We also note that saddle points are a known problem in optimization [4]. In similar style to [3], we believe an analysis using Lemma 3.2 to characterize the ratio of hidden symmetry induced critical spaces and global minima spaces as we overparameterize can provide insight on scaling benefits of equivariant vs. unconstrained models.

## References

[1] Fukumizu, Kenji, and Shun-ichi Amari. "Local minima and plateaus in hierarchical structures of multilayer perceptrons."

[2] Pacini, Marco, Xiaowen Dong, Bruno Lepri, and Gabriele Santin. "Separation Power of Equivariant Neural Networks."

[3] Simsek, Berfin, François Ged, Arthur Jacot, Francesco Spadaro, Clément Hongler, Wulfram Gerstner, and Johanni Brea. "Geometry of the loss landscape in overparameterized neural networks: Symmetries and invariances."

[4] Dauphin, Yann N., Razvan Pascanu, Caglar Gulcehre, Kyunghyun Cho, Surya Ganguli, and Yoshua Bengio. "Identifying and attacking the saddle point problem in high-dimensional non-convex optimization."

---

### Author Response · Authors · 2025-08-08
**Hostile Comments from the Area Chair**

Dear Reviewers, Senior Area Chairs, and Program Chairs,

We would first like to thank the PCs and senior ACs for overseeing the entire review process. We would also like to thank the reviewers for their extensive and helpful feedback on our work.

We are grateful that our assigned AC has spent additional time taking a detailed look at our manuscript. We genuinely value constructive feedback.

However, the feedback received from the AC seems to have major misunderstandings of our work. Further, we are surprised and concerned by the hostility of the review, especially in their most recent comment.

We would greatly appreciate it if the reviewers and senior ACs could provide their input on this situation and the presented evidence.

Best,

Authors

---

> ### Comment · Reviewer_AZWo · 2025-08-09
>
> I agree with the authors' concerns. The last comment of the area chair was very unfair towards your contribution, and is overexaggurating and/or misrepresenting relatively minor weaknesses/inaccuracies of the work. An additional discussion thread has been opened, where I think that the authors have not been made readers, and I have expressed this view also in that thread.
>
> I support the senior area chairs taking a look at this case.

---

### Note · Authors · 2025-08-15

We would like to once again thank everyone for their time in evaluating our work. We are grateful for the numerous detailed and thoughtful comments. Here, we summarize the main changes we plan to add if accepted.

## Important changes
* Add the missing assumption we forgot to include for Theorem 3.3
* Discuss the satisfiability of each assumption as summarized in our general response
     * Emphasize how the co-dimension assumption limits direct applicability
     * Relax non-degenerate Hessian assumption from all critical points to only the point in the hidden fixed point space as suggested by reviewer jLby
* Discuss how Lemma 3.2 can be used for future work as summarized in our general response

## Minor changes
* As suggested by reviewer Xb7d, double check that all symbols are well defined, even if their meaning can be inferred. Add a summary of our notation and common meanings of the symbols in the appendix
* Clarify we will only consider parameter symmetries with linear action in this paper after line 131 (note this includes most common parameter symmetries such as permutation of neurons)
* Explicitly describe the action of the parameter symmetry of permuting neurons (see comments to reviewer AZWo)
* Simplify line 568 as suggested by AC mkg8
* Add a brief discussion of how our insights to the loss landscape affects different optimizers as suggested by reviewer Ppai
* Add a separate related works section briefly discussing some of the works brought up by the reviewers and AC with a more in depth discussion (as found in the rebuttals) in the appendix
* Add some figures to illustrate the hidden symmetry idea. Further, if space allows, provide proof sketches as suggested by reviewer cpcL

Lastly, we again want to emphasize the key messages of our work

## Key messages
* By viewing a given architecture as a constrained version of a larger function class, we prove that the previously hidden parameter symmetries of the larger class can imply the existence of critical points and even spurious minima for the constrained network
* Hidden symmetries induce multiple symmetrically related but different constrained subspaces. Empirically, we find a spurious minimum in an equivariantly constrained subspace is often a saddle in the unconstrained space and escapes to a better minimum in a symmetrically related subspace.
* To achieve effective relaxation of equivariant models, one may have to rethink our fixed choice of group representation in the hidden layers

---

### Decision · Program_Chairs · 2025-09-17

**Decision:**

Accept (poster)

**Comment:**

The paper considers the following setting:

- $\mathcal{F}\_\theta = \\{f\_\theta : X \to Y \mid \theta \in \Theta\\}$ is a class of functions over the parameter space $\Theta$, and $L:\Theta \to \mathbb{R}$ a loss function.

- $H\le P$, where $P$ is a group of global symmetries (i.e., $f_\theta = f_{g\theta}$ for all $g \in P$ and $\theta \in \Theta$).

- $\mathcal{F}_{\tilde{\Theta}}$ is a class of functions corresponding to $\tilde{\Theta} \subset \Theta$, interpreted as the usual restriction to weights that yield equivariant layers.

The main result concerns conditions under which two minima may be found on $\tilde{\Theta}$.
Specifically, assuming that: the loss function is $C^1$, there exists a minimum
 $\theta_1 \notin\Theta\_H\cap \tilde{\Theta}$ for $L|\_{\tilde{\Theta}}$,
$\mathrm{codim}(\Theta_H \cap \tilde{\Theta}) = 1$, there exists a constant $C$ such that
for any direction $\hat{r} \in \tilde{\Theta}$ one has $\hat{r} \cdot \nabla L(C\hat{r}) > 0$,
Hessians of critical points are nonsingular, and $(\Theta_H \cap \tilde{\Theta}) \oplus (\Theta_H^\perp \cap \tilde{\Theta}) = \tilde{\Theta},$
then there exists a second minimum $\theta_2 \neq \theta_1$ for $L|_{\tilde{\Theta}}$.
The result is subsequently specialized to cases involving permutation representations.
The experimental section presents an experiment in the teacher–student setting, illustrating a scenario where the theorem’s implication appears to hold without satisfying the stated assumptions.

Several issues were raised and communicated to the authors relating to core definitions, the formulation, proofs (Lemma 3.2 and Theorem 3.3) and scope of the main result, as well as its empirical validation. In particular, reviewers expressed concerns regarding the practical applicability of the results, citing the codimension and nondegeneracy assumptions as well as the numerical experiments. During the discussion, it was remarked that in the experimental section several assumptions underlying the theoretical result were not verified, and some were violated, leaving the empirical verification of the result open. It was further noted that the minima observed had been reported and studied in prior work. The authors have addressed these issues to the satisfaction of the reviewers, who appreciated the importance of the problem concerning the optimization of equivariant networks, the stated result regarding the existence of an additional minimum in the subspace $\tilde{\Theta}$, and the empirical results on the relaxation of layer equivariance.